# Experiments on Flexible Filaments in Air Flow for Aeroelasticity and Fluid-Structure Interaction Models Validation

**Jorge Silva-Leon [1] and Andrea Cioncolini [2],***

[1]   Escuela Superior Politécnica del Litoral, ESPOL, Facultad de Ingeniería en Mecánica y Ciencias de la Producción, Campus Gustavo Galindo Km 30.5 Vía Perimetral, Guayaquil P.O. Box 09-01-5863, Ecuador; jfsilva@espol.edu.ec

[2]   Department of Mechanical, Aerospace and Civil Engineering, University of Manchester, George Begg Building, Sackville Street, Manchester M1 3BB, UK

*   Correspondence: andrea.cioncolini@manchester.ac.uk

**Abstract:** Several problems in science and engineering are characterized by the interaction between fluid flows and deformable structures. Due to their complex and multidisciplinary nature, these problems cannot normally be solved analytically and experiments are frequently of limited scope, so that numerical simulations represent the main analysis tool. Key to the advancement of numerical methods is the availability of experimental test cases for validation. This paper presents results of an experiment specifically designed for the validation of numerical methods for aeroelasticity and fluid-structure interaction problems. Flexible filaments of rectangular cross-section and various lengths were exposed to air flow of moderate Reynolds number, corresponding to laminar and mildly turbulent flow conditions. Experiments were conducted in a wind tunnel, and the flexible filaments dynamics was recorded via fast video imaging. The structural response of the filaments included static reconfiguration, small-amplitude vibration, large-amplitude limit-cycle periodic oscillation, and large-amplitude non-periodic motion. The present experimental setup was designed to incorporate a rich fluid-structure interaction physics within a relatively simple configuration without mimicking any specific structure, so that the results presented herein can be valuable for models validation in aeroelasticity and also fluid-structure interaction applications.

**Keywords:** experiment; benchmark; validation; aeroelasticity; fluid-structure interaction; flexible; filament; ribbon; string

## 1. Introduction

Fluid-structure interaction (FSI) problems, where a fluid flow and a movable or deformable structure dynamically interact, are relevant in several fields of engineering including aeroelasticity, biomechanics, flow control, and energy harvesting. Examples include aircraft wing design [1,2], flapping wing propulsion [3–5], flexible turbomachinery [6,7], cardiovascular medicine [8–11], swimming micro-organisms [12,13], piezoelectric wind energy harvesting [14,15], and several more. A flexible structure exposed to fluid flow deforms owing to the fluid force acting along its surface. When the deformation of the structure is large enough to affect the flow field, the resulting FSI problem is a coupled, non-linear multi-physics problem where the flow and the structure dynamically interact and modulate each other.

Despite their practical relevance, a comprehensive treatment of FSI problems remains a challenge due to their intrinsic complexity and multidisciplinary nature. FSI problems are typically too complex to solve analytically, and are therefore analyzed by means of experiments and numerical simulations.

Frequently, conducting experiments at operating conditions representative of actual applications may be challenging or impractical. In these cases, numerical modelling may be used as the main design and analysis tool to investigate the fundamental physics of FSI problems. The development of numerical methods for FSI problems has been an active area of research over the last decades, and several numerical procedures have been proposed to account for large structural deformations and faithfully reproduce the coupling between the structure and the fluid [16–19]. The effectiveness of numerical FSI methods is normally assessed via verification and validation. Whilst the verification is carried out by comparing the simulations with synthetic data generated with numerical experiments (see, e.g., [20–24]), the validation relies on comparing the simulations with experimental test cases, i.e., data originated from physical experiments. In order to be informative and, at the same time, minimize the computational burden, the experimental test cases used to validate numerical FSI methods do not normally mimic any realistic structure. Instead, these experimental test cases are typically designed to incorporate a rich fluid-structure interaction physics within a relatively simple configuration. For illustrative purposes, a non-exhaustive selection of popular experimental FSI test cases is provided in Table 1.

**Table 1.** Experimental FSI validation test cases.

| Reference | Fluid | Structure | Motion |
|---|---|---|---|
| Pereira Gomez et al. [25] | Polyethylene glycol syrup in laminar flow | Flexible metal plate with rear mass at the trailing edge, clamped behind a rigid circular cylinder free to rotate around its axis, oriented in cross-flow | 2D |
| Pereira Gomes and Lienhart [26] | Water and polyethylene glycol syrup in laminar and turbulent flow | Flexible metal plate with rear mass at the trailing edge, clamped behind a rigid cylindrical body (circular or rectangular cross section) oriented in cross-flow | 2D |
| Kalmbach and Breuer [27] | Water in turbulent flow | Flexible rubber membrane with rear mass at the trailing edge, clamped behind a rigid and fixed circular cylinder oriented in cross-flow | 2D |
| De Nayer et al. [28] | Water in turbulent flow | Flexible rubber membrane clamped behind fixed rigid circular cylinder oriented in cross-flow | 2D/3D |
| Hessenthaler et al. [29] | Aqueous glycerol solution in laminar flow | Flexible cantilever-beam plate in merging flow from two inlets | 3D |
| This study | Air in laminar and turbulent flow | Flexible cantilever-beam filaments of variable length in uniform flow | 3D |

As can be noted in Table 1, the experimental setups comprise an elastic structure of simple geometry, such as a plate or a membrane, which undergoes large deformations with moderate motion frequency whilst interacting with a flow of moderate Reynolds number, so that the complication of simulating highly turbulent flows is avoided. Sometimes, the structure dynamics is restricted to two-dimensional, so that the validation can be achieved by means of two-dimensional numerical simulations, which are less demanding than three-dimensional simulations in terms of run-time and computational resources. In addition to the structural dynamics, which is always resolved in validation experiments, when practical also the flow field is measured. Faithfully resolving this latter is not always feasible, however, particularly when the structural deformation is three-dimensional, so that flow field measurements, when provided, are frequently of low resolution and of limited scope.

With the rapid development of numerical FSI methods, the demand for validation test cases increases. The objective of this work is to contribute one such case. In particular, we tested six flexible filaments of rectangular cross-section and varying length, which were exposed to air flow of moderate Reynolds number corresponding to laminar and mildly turbulent flow conditions. In order to explore a wider range of structural responses during the tests, besides varying the air flow velocity we also varied the length of the elastic filaments. This was instrumental to observing structural responses as diverse

as: (1) static reconfiguration, (2) small-amplitude vibration, (3) limit-cycle periodic oscillations, and (4) non-periodic oscillations. The dynamics of the flexible filaments was generally three-dimensional, though a two-dimensional structural response was sometimes observed during limit-cycle periodic oscillations. Even though the focus here is clearly on three-dimensional large deformations of flexible structures in air flow, the practical relevance of the work goes beyond aeroelasticity applications, and the present results can be of interest for FSI applications in general.

The rest of this paper is organized as follows: the test set-up, the flexible filaments characterization, the flow characterization and the experimental methodology are presented in Section 2, whilst the measured results are presented and discussed in Section 3.

## 2. Materials and Methods

### 2.1. Flexible Filament Manufacturing and Characterization

The flexible filament was manufactured with commercial silicone rubber (density $1.00 \pm 0.05$ g/cm$^3$) using an additive manufacturing system (3D-Bioplotter by EnvisionTEC, https://envisiontec.com) following a rectilinear path during printing to avoid any curvature or deformation, and therefore produce a straight filament of uniform cross section and smooth surface finishing. The filament had rectangular cross section with width $w$ of $2.00 \pm 0.05$ mm and height $h$ of $0.40 \pm 0.05$ mm (measured with a digital caliper), and a linear mass density of $0.80 \pm 0.16$ g/m. Note that, in order to avoid the large error that would have arisen from measuring directly the mass of the filament (which was on the order of 0.1 g), the density of the silicon rubber provided above was deduced from measuring the mass of a bigger chunk of silicon rubber. The linear mass density of the filament provided above was therefore calculated as the product of the silicon rubber density times the width and height of the filament cross-section. One portion of the filament was used for the mechanical characterization described below, whilst another portion was used to realize the test piece for the FSI experiments (described in Section 2.2).

The mechanical behavior of the filament was characterized with uniaxial tensile tests. The results are provided in raw format in Figure 1a as filament elongation (measurement accuracy $\pm 0.5$ mm) versus applied force (measurement accuracy $\pm 1\%$), whilst the corresponding stress-strain curve is provided in Figure 1b. The tensile strain $\epsilon$ and the tensile stress $\sigma$ are calculated as indicated in Equations (1) and (2), respectively:

$$\epsilon = \frac{\Delta L}{L_0} = \frac{L - L_0}{L_0} \tag{1}$$

$$\sigma = \frac{F}{A} = \frac{FL}{A_0 L_0} \tag{2}$$

where $\Delta L$ is the filament elongation, $L$ is the length of the filament when loaded, $L_0$ is the initial length of the filament (note that the filament used for the tensile test had an initial length of $52.0 \pm 0.5$ mm), $F$ is the applied force, and $A$ is the area of the cross-section of the filament. As can be noted in Equation (2), the tensile stress was calculated by assuming incompressible deformation, which is an acceptable approximation for silicone rubber [29]. No permanent deformation was observed after loading and the data in Figure 1b are linearly correlated, therefore indicating a linear-elastic behavior of the flexible filament during the tensile tests. The experimental uncertainties, estimated with standard single-sample error propagation [30], were on the order of $\pm 2\%$ for the tensile strain, and $\pm 10\%$ for the tensile stress. The corresponding Young modulus of the filament, deduced from the slope of the fitting line in Figure 1b, is $E = 757 \pm 91$ kPa.

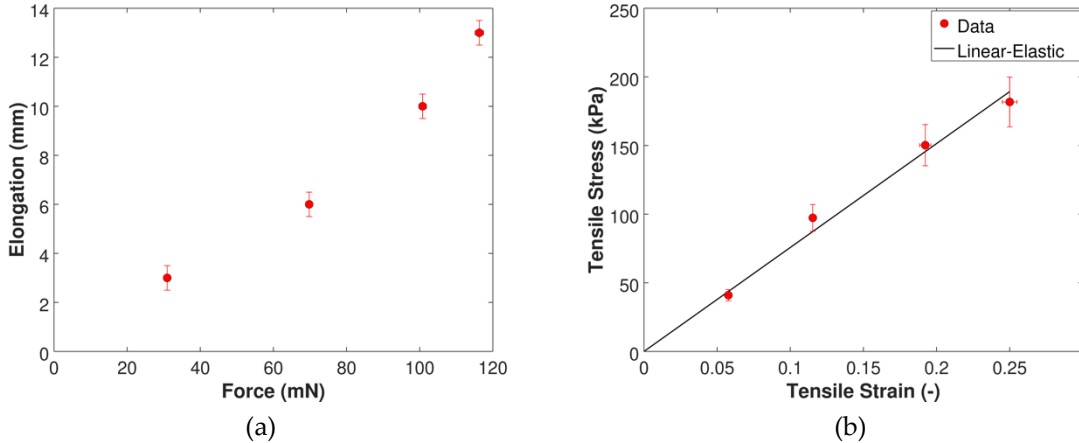

**Figure 1.** Mechanical characterization of the flexible filaments. (**a**) Uniaxial tensile tests results; (**b**) Stress-strain curve.

## 2.2. Test Piece Description and Preliminary Characterization

As schematically shown in Figure 2a, the test piece for the FSI experiments comprises the flexible filament, a support tube, and a support plate. The support tube, a rigid stainless-steel circular tube with external diameter of 2.40 ± 0.05 mm and length of 140 ± 0.5 mm, was rigidly connected to the support plate so that the test piece could be introduced from the top inside the wind tunnel for testing, as shown in Figure 2c. The filament extremity was introduced inside the support tube and then glued, so as to realize a cantilever boundary condition at the connection between the filament and the tube. The length of the support tube was selected to place the filament in the middle of the wind tunnel cross-section. During the tests, the filament was always oriented with the longer side of the rectangular cross section facing the flow, as schematically shown in Figure 2b. The flexible filament protruding from the support tube had an initial length $L$ of 60 mm. The filament was progressively shortened during the experiments, and tests were carried out for six different lengths: 60 mm, 50 mm, 40 mm, 30 mm, 20 mm, and 10 mm. These are referred to, in the following, as Filament 1 (60 mm) through Filament 6 (10 mm), as indicated in Table 2.

**Table 2.** Flexible filaments used in the present FSI experiments.

| Filament No. | $L$ (mm) | $f_1$ (Hz) | $\zeta_1$ (-) |
|:---:|:---:|:---:|:---:|
| 1 | 60.0 ± 0.5 | 2.5 ± 0.1 | 0.012 ± 0.002 |
| 2 | 50.0 ± 0.5 | 2.8 ± 0.1 | 0.015 ± 0.002 |
| 3 | 40.0 ± 0.5 | 3.3 ± 0.2 | 0.017 ± 0.002 |
| 4 | 30.0 ± 0.5 | 3.9 ± 0.3 | 0.021 ± 0.003 |
| 5 | 20.0 ± 0.5 | 6.3 ± 1.0 | 0.028 ± 0.004 |
| 6 | 10.0 ± 0.5 | 16 ± 3 | 0.035 ± 0.005 |

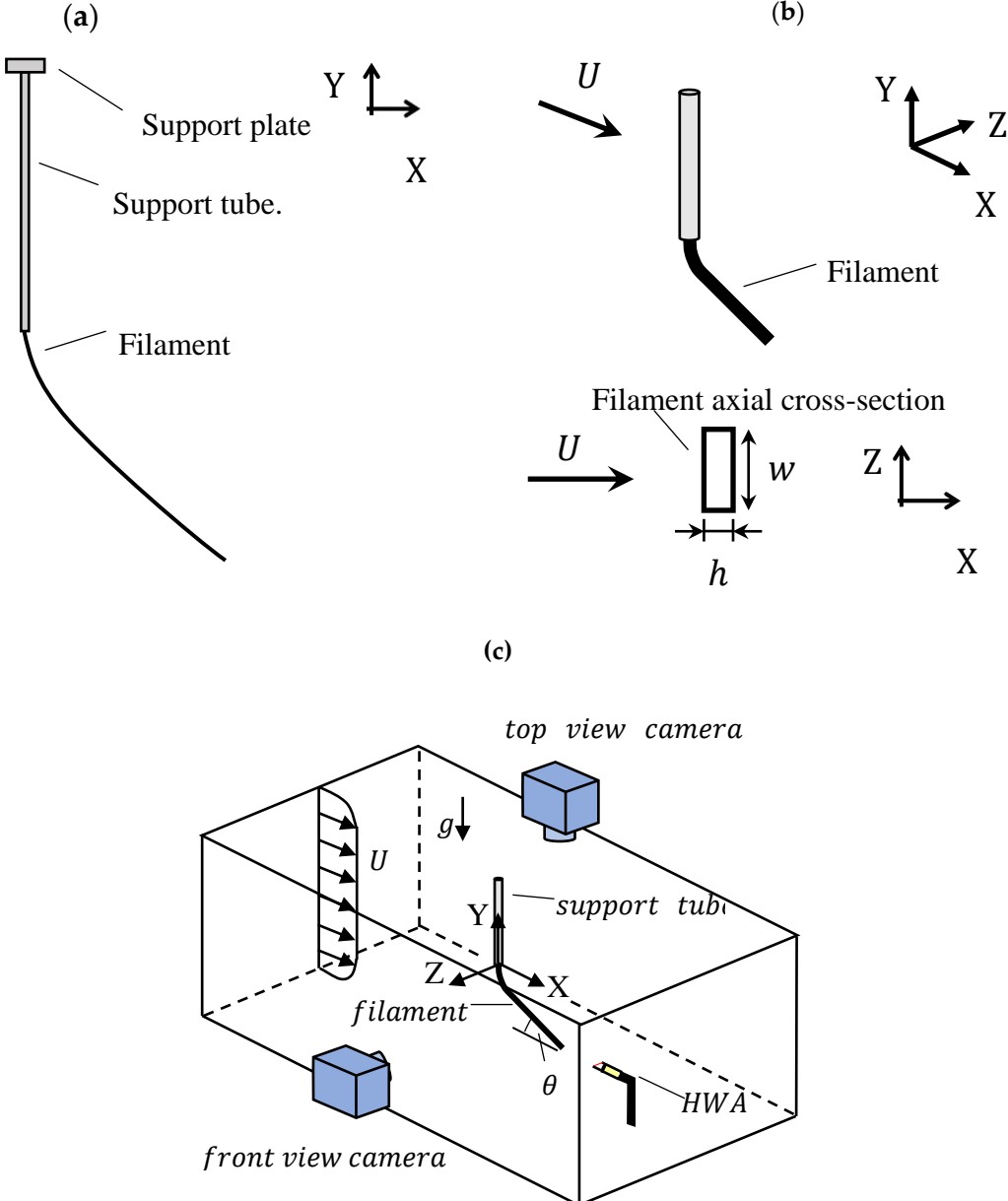

**Figure 2.** (**a**) Schematic representation of the test piece; (**b**) Filament orientation during the tests in the wind tunnel; (**c**) Schematic representation of the experimental wind tunnel setup.

Before running the FSI tests in the wind tunnel, the filaments were preliminary characterized by measuring their natural vibration frequencies and damping ratios. The results are provided in Table 2. In particular, first-mode natural vibration frequencies $f_1$ were measured in forced vibration shaker tests under single-frequency excitation. The test setup included an electromagnetic shaker (of in-house design and construction) with control signal provided by a signal generator operated in sine wave mode with frequency resolution of 0.1 Hz. During the tests, the filaments hang vertically with the top extreme fixed to the shaker. Following common practice, the amplitude of response of the filament was recorded (using a Panasonic Lumix DMC-FZ200 digital camera) as a function of the excitation frequency, and the natural vibration frequency was identified as the peak in the response (experimental uncertainty deduced from the full-width at half maximum of the peak in the amplitude response). On the other hand, first-mode damping ratios $\zeta_1$ were deduced from free vibration tests in stagnant air. Starting with the filament hanging vertically in equilibrium with the top extreme fixed, the filament free-end was manually displaced (displacement small enough to trigger a mode-1

response). The filament free-end was then released, and the free vibration of the filament was recorded (using a Panasonic Lumix DMC-FZ200 digital camera). Following common practice, the damping ratio was evaluated from the logarithmic decrement of the envelope of the displacement time-series (experimental uncertainty deduced as standard deviation from repeated measurements). Natural vibration frequencies of the filaments are presented in Figure 3a together with the predictions of Equations (3) and (4):

$$f_{Beam} = \frac{1.875^2}{2\pi} \sqrt{\frac{E\,I}{m\,L^4}} \tag{3}$$

$$f_{String} = \frac{2.4048}{4\pi} \sqrt{\frac{g}{L}} \tag{4}$$

where $E$ is the Young modulus of the filament, $m$ is the total linear mass density of the filament (since the density of the filament is 3 orders of magnitude larger than the density of air, the added mass is negligible in the present case), $g$ is the acceleration of gravity, and $I$ is the second area moment of the cross-section of the filament with respect to the axis aligned with the longer side ($z$-axis in Figure 2b):

$$I = \frac{wh^3}{12} \tag{5}$$

In particular, Equation (3) predicts the first-mode natural vibration frequency of a cantilevered elastic beam according to standard Euler-Bernoulli beam theory, whereas Equation (4) predicts the first-mode natural vibration frequency of a one-dimensional elastic and inextensible hanging string [31]. As can be noted in Figure 3a, whilst the natural vibration frequencies of the shorter filaments (Filaments 5 and 6) agree with Equation (3), those of the longer filaments (Filaments 1 through 4) agree with Equation (4). This indicates that, from a structural point of view, the shorter filaments behave as elastic beams, whereas the longer ones behave as elastic strings. Even though this conclusion, strictly speaking, is only valid for the forced vibration tests discussed here, the FSI tests discussed later provide similar results: the structural response of the filaments is modulated by their length. This highlights the usefulness of using the filament length as control parameter during the experiments to explore different structural responses. As can be noted in Figure 3b, the damping ratio decreases when increasing the filament length, confirming previous observations with flexible filaments of circular cross section [32,33].

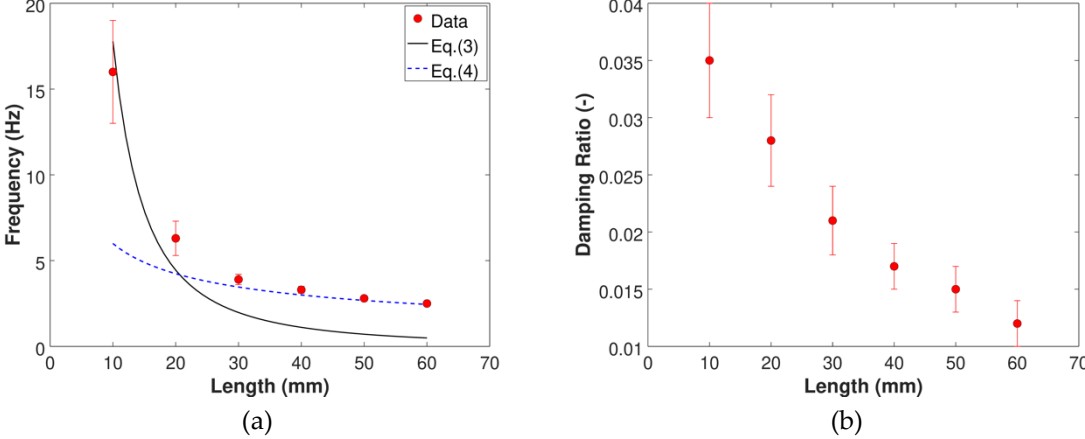

**Figure 3.** Preliminary characterization of the flexible filaments. (**a**) First-mode natural vibration frequency; (**b**) First-mode damping ratio in stagnant air.

### 2.3. Wind Tunnel Flow Characterization

The experiments were performed in a horizontal-axis commercial wind tunnel (by Armfield Limited, Ringwood, UK (armfieldonline.com)) of octagonal cross-section with height and width of

300 mm, operated with air at ambient conditions (air pressure and temperature during the experiments were 101 ± 1 kPa and 293 ± 2 K, respectively). The blockage ratio associated with the present test piece was on the order of 0.5%, so that wind tunnel flow confinement effects can be ignored. The free-stream flow velocity was measured (to within ± 2% accuracy) with a hot-wire anemometer (by Dantec Dynamics, Bristol, UK (www.dantecdynamics.com); probe type 55P15: 5 μm diameter tungsten wire of 2 mm length), calibrated prior to the tests and operated in constant temperature mode with a sampling frequency of 10 kHz. As shown schematically in Figure 2c, the hot-wire anemometer (HWA) was located downstream of the test piece in the same vertical plane of the filament but at a lower vertical elevation, so as to avoid any interference between the anemometer and the wake of the filament.

　　The uniformity of the free-stream velocity profile and the extension of the boundary layer in the wind tunnel were assessed before introducing the flexible filaments in the tunnel for testing. The boundary layer thickness was of about 5 mm at the lowest wind speed considered, whilst the velocity profile (excluding the boundary layer) was uniform to within ± 2%, i.e., velocity variations were within the present experimental resolution. This assures that the flexible filaments were always exposed to a fully-developed velocity profile during the tests. The free-stream wind tunnel flow was further characterized by estimating the streamwise component of the turbulence intensity $Tu$ and the streamwise macro $\Lambda$ and micro $\lambda$ turbulence length scales [34]:

$$Tu = \frac{\sqrt{\overline{u^2}}}{U} \tag{6}$$

$$\Lambda = \left[ \frac{E(f)U}{4\overline{u^2}} \right]_{f \to 0} \tag{7}$$

$$\lambda = \left[ \frac{2\pi^2}{U^2\overline{u^2}} \int_0^\infty f^2 E(f) df \right]^{-2} \tag{8}$$

where $\overline{u^2}$ is the mean square fluctuating velocity, $U$ is the mean flow velocity, and $E(f)$ is the energy spectrum of the velocity signal as function of the frequency $f$. Equations (7) and (8), in particular, are valid under the assumption of homogeneous and isotropic turbulence, which is normally acceptable for wind tunnel flow experiments [34]. As is well known, the turbulence intensity measures the relative intensity of the velocity fluctuation. On the other hand, the turbulence macroscale can be considered as a measure of the largest eddy size in the flow, whilst the turbulence microscale can be considered a measure of the smallest eddies in the flow, which are responsible for the dissipation of turbulence energy. Turbulence intensity and length scales estimates are presented as functions of the wind speed in Figure 4. As can be noted, the turbulence intensity is mild and ranges between 1% and 2.5%, whilst the macro and micro length scales range between 1−30 mm and 0.02−0.38 mm, respectively. The estimates provided in Figure 4 were generated from five-seconds long hot-wire velocity measurements of the flow, sampled at 10 kHz. The free-stream flow is not further characterized at this stage, but raw free-stream flow data are included as Supplementary Materials for future reference and further analysis.

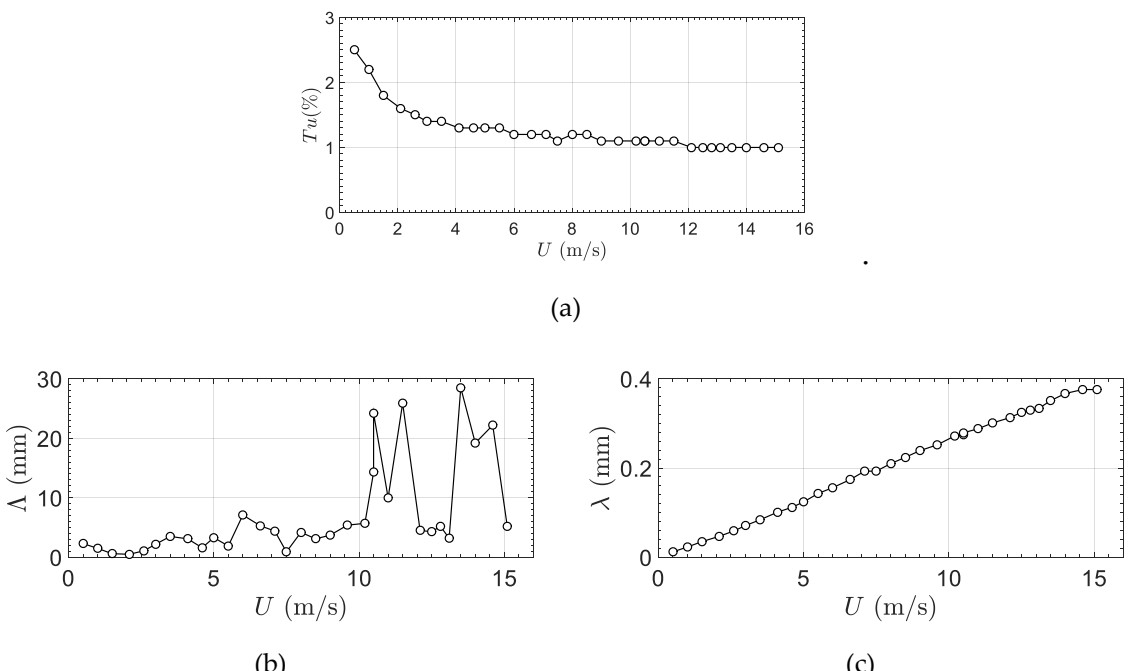

(a)

(b)

(c)

**Figure 4.** Preliminary characterization of the free-stream wind tunnel flow. (**a**) Streamwise turbulence intensity; (**b**) Streamwise turbulence macroscale; (**c**) Streamwise turbulence microscale.

### 2.4. Experimental Procedure

During the FSI tests, the free-stream flow velocity was gradually and stepwise varied between 1 m/s and 15 m/s, corresponding to a Reynolds number in the range of 133−2027. The Reynolds number, in particular, is based on the filament width $w$:

$$Re = \frac{\rho U w}{\mu} \tag{9}$$

where $\rho$ and $\mu$ are the air density and viscosity. With cylinders in cross-flow, the vortex street becomes turbulent for Reynolds numbers above about $Re \approx 200 - 300$ [35]. The Reynolds number range explored here, therefore, covers laminar (flow velocity up to ~1.5 m/s), transitional (flow velocity from ~1.5 m/s up to ~2.5 m/s), and mildly turbulent (flow velocity above ~2.5 m/s) flow conditions. Measurements were taken for both increasing and decreasing flow velocity, observing no hysteresis in the response of the filaments.

The motion of the filaments was recorded simultaneously in the horizontal and vertical planes using two synchronized digital cameras (Panasonic Lumix DMC-FZ200, recording frequency: 200 frames per second, image resolution: 480 × 640 pixels), located at the front and at the top of the wind tunnel as shown in Figure 2c. The digital cameras provided a spatial resolution of 0.20 mm/pixel, which is appropriate to resolve the large displacements of interest here. The videos (15 s recording) were synchronized and post-processed with the Image Processing Toolbox of MATLAB (version R2015a), using a tracking methodology previously developed for flexible fluid-structure interaction and flow-induced vibration applications [32,33,36–38].

Representative envelopes of motion for Filament 1 at three different Reynolds numbers ($Re = 202$, $Re = 410$, and $Re = 804$) are presented in Figure 5. Except at the lowest Reynolds number value ($Re = 202$), when the motion of the filament is two-dimensional and confined to the vertical plane, the motion of the filament is clearly three-dimensional, with significant displacements in both the vertical and horizontal planes. Not surprisingly, the free-end is the point along the filament that experiences the largest displacement. As shown in previous research on flexible filaments in cross-flow [32,33], the dynamics of the filament free-end is representative of the dynamics of any point along the flexible

filament, meaning that the dynamics of any point along the flexible filament is qualitatively similar to the dynamics of the filament free-end, despite the different amplitude of motion. In the present study, therefore, the response of the filaments was characterized by analyzing the dynamics of the filament free-end, which experiences the largest displacement and therefore maximizes the signal-to-noise ratio. On account of the filament motion being mostly three-dimensional, the filament free-end dynamics was characterized by analyzing the displacement time-series $A_y(t)$ and $A_z(t)$ recorded in the vertical plane (front view in Figure 2c) and horizontal plane (top view in Figure 2c), and by analyzing the filament free-end total displacement time-series $A_{tot}(t)$:

$$A_{tot}(t) = \sqrt{A_y(t)^2 + A_z(t)^2} \tag{10}$$

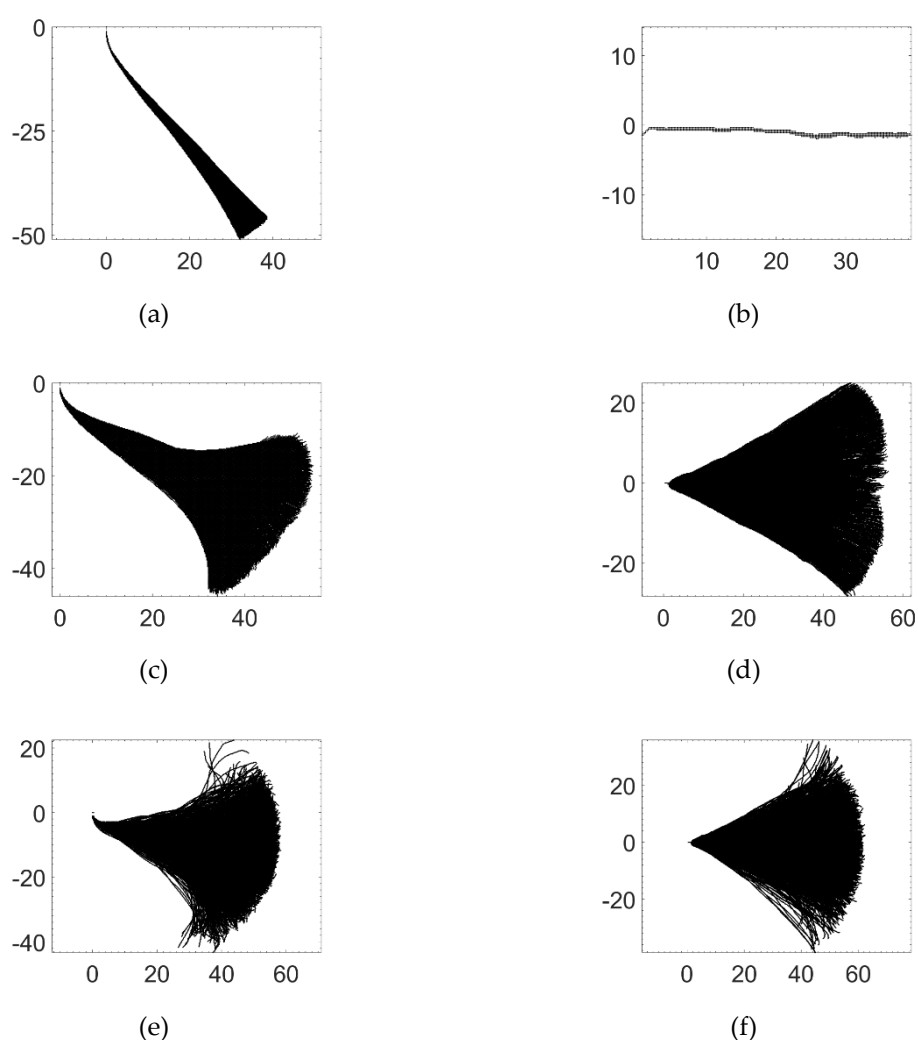

**Figure 5.** Representative envelopes of motion for Filament 1 in the vertical plane (left) and horizontal plane (right) at three Reynolds numbers (top, middle and bottom). (**a**) Vertical plane, $Re = 202$; (**b**) Horizontal plane, $Re = 202$; (**c**) Vertical plane, $Re = 410$; (**d**) Horizontal plane, $Re = 410$; (**e**) Vertical plane, $Re = 804$; (**f**) Horizontal plane, $Re = 804$ (*x*-axis and *y*-axis scales are in mm).

The total displacement combines the vertical and horizontal motions of the filament free-end, and can therefore be used to characterize both planar and three-dimensional dynamics. In particular, the recorded time-series were not filtered prior to the analysis. Flapping frequencies in the vertical and horizontal planes were identified as the dominant frequencies in the power spectral densities

(computed using the Welch method [39] and MATLAB built-in functions) of the corresponding displacement time-series.

The total displacement time-series was used to compute the autocorrelation function and to reconstruct the system trajectory in phase-space, which together provide a thorough characterization of the filament free-end dynamics. Whilst the autocorrelation function was computed using MATLAB built-in functions, the trajectory of the system in phase-space (i.e., the system attractor) was reconstructed using the delayed vector method [40]: a methodology developed for nonlinear time-series analysis that is particularly useful for fluid-structure interaction problems. The topology of the reconstructed attractor of the filament free-end, in fact, gives a useful qualitative characterization of the filament dynamics, which completes and corroborates the quantitative information provided by the displacements and flapping frequency. The concept of phase-space representation, rather than a classic analysis in time or frequency domain, is the key point in nonlinear time-series analysis. The topology of the trajectory in phase-space of a nonlinear system, in fact, provides important qualitative information regarding the fundamental dynamics of the systems being investigated. The problem is that, in experimental studies, one normally observes a time-series of scalar measurements of some quantity that depends on the current state of the system, and not the trajectory of the system in phase-space. The delayed vectors method [40] allows reconstructing the trajectory of the system in phase-space from a time-series of scalar measurements, and can therefore be used in experimental studies such as the present one where only time-series of scalar measurements of some quantity (the instantaneous total displacement in the present case) are available. In the present case, the trajectory in phase-space was reconstructed by plotting the time-delayed instantaneous total displacement $A_{tot}(t + \tau)$ versus $A_{tot}(t)$. If the delay $\tau$ is chosen properly (on the order of 5-30 ms in the present case, the lower the delay the higher the flow velocity and Reynolds number), then the topology of the reconstructed attractor is representative of the underlying system dynamics [40].

As noted previously, experimental FSI test cases should always include the structural dynamics and, when feasible, also the measurement of the flow field. Measuring the flow field in the present case would require a three-dimensional flow visualization with sub-millimeter space resolution and high-frequency (on the order of several kHz) time resolution. Even for the simplest test case documented here, which corresponds to Filament 6 statically reconfigured at a flow velocity of 1 m/s (corresponding to a Reynolds number of $Re = 133$), measuring the flow field would require the resolution of a vortex street where vortices of sub-millimetric size are shed at a frequency of about 100 Hz [41]. As previously noted, in FSI test cases it is normally preferred to have flexible structures which undergo large deformations with moderate motion frequency whilst interacting with a flow of moderate Reynolds number, thereby avoiding the complications of simulating highly turbulent flows. The small size and high flexibility of the present filaments were instrumental in achieving large displacements and relatively small oscillation frequencies and, at the same time, keep the Reynolds number small. Unfortunately, the details of the flow field scale with the size of the structure, so that the smaller the structure the smaller the space resolution that is needed to faithfully resolve the flow field. In the present case, the requirements for a faithful flow field visualization were beyond our experimental capabilities, and the flow field was therefore not measured.

## 3. Results and Discussion

The results for Filament 1 are presented in Figures 6–9. In particular, displacements and flapping frequencies in the horizontal and vertical planes are presented in Figure 6 as functions of the Reynolds number, whilst the trajectory of the filament free-end (as seen by an observed located downstream of the filament and facing the flow), the autocorrelation function and the reconstructed attractor (computed from the total displacement of the filament free-end) are provided, for all Reynolds number values tested, in Figures 7–9. Each data point presented in the paper has been generated from averaging one video recording of 15 seconds, corresponding to 3000 frames. Selected data points were repeated, showing good repeatability.

It is evident that the filament response gradually changes as the Reynolds number is progressively increased. For low values of the Reynolds number ($Re \lesssim 333$) the filament motion is small-amplitude and mostly confined to the vertical plane, with a rapidly decaying autocorrelation function and a blob-like attractor. These are typical features of a small-amplitude vibration: a random and not self-sustained motion confined around the equilibrium position corresponding to the statically reconfigured filament.

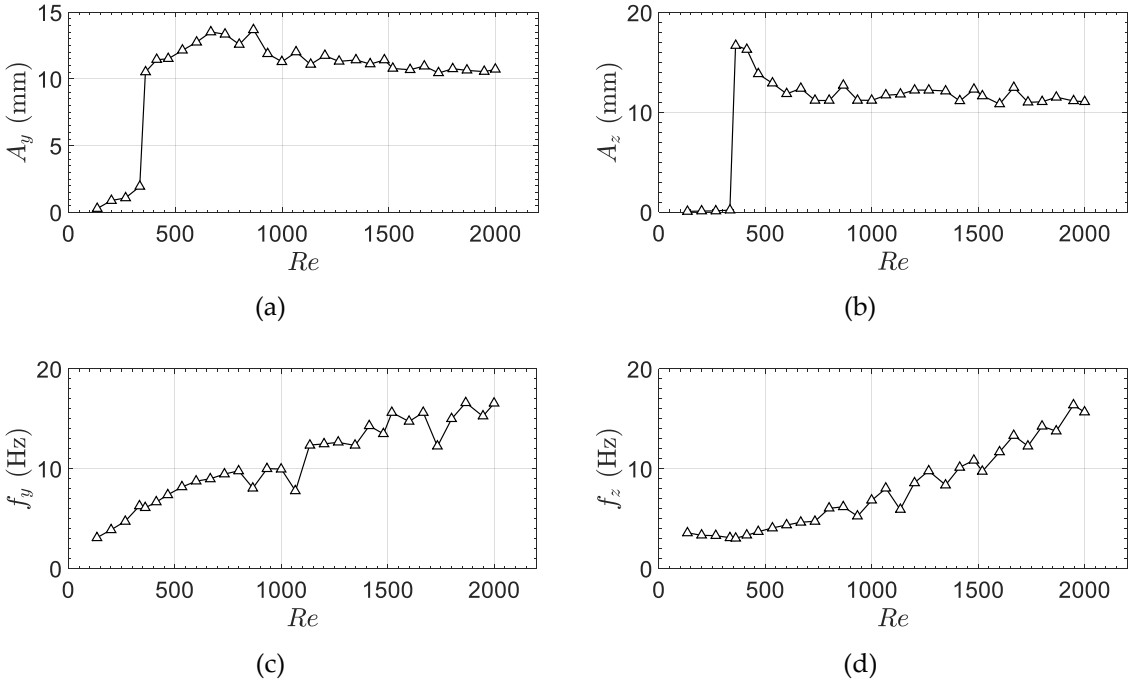

**Figure 6.** Displacements and flapping frequencies measured for Filament 1. (**a**) Displacement in the vertical plane; (**b**) Displacement in the horizontal plane; (**c**) Frequency in the vertical plane; (**d**) Frequency in the horizontal plane.

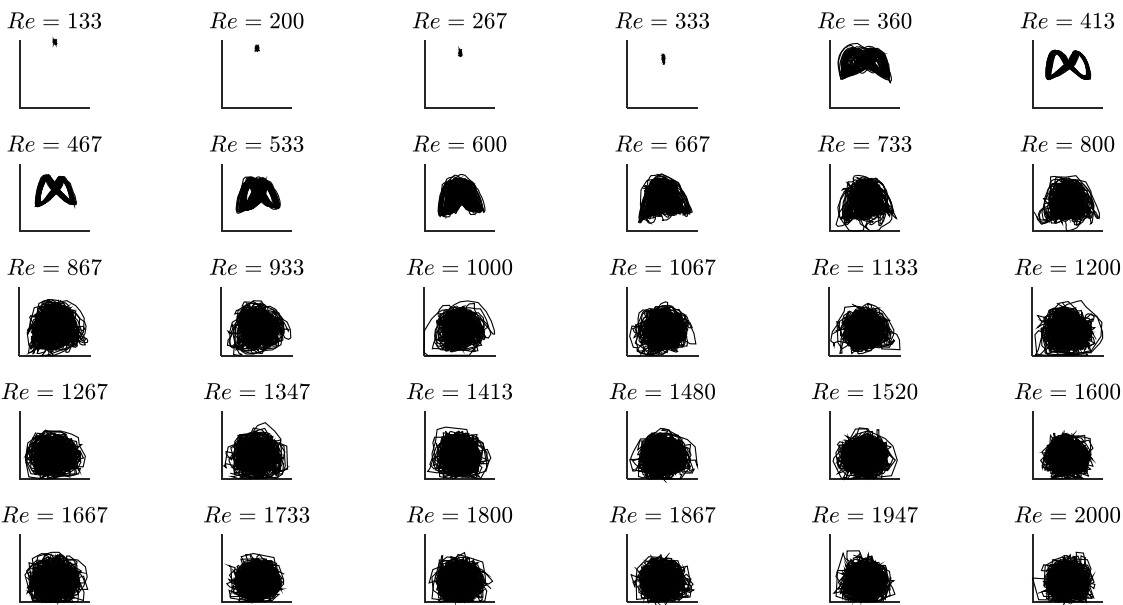

**Figure 7.** Trajectory of the free-end of Filament 1 as seen by an observer located downstream of the filament and facing the flow (Y-Z plane in Figure 2c).

For Reynolds numbers beyond about $Re = 360$, the filament motion becomes large-amplitude and three-dimensional, with displacements in the horizonal and vertical planes of comparable magnitude. For Reynolds numbers in the range of $413 - 533$, in particular, the autocorrelation function becomes periodic and slowly decaying and the attractor is ring-like, indicating that the filament motion is a large-amplitude limit-cycle oscillation where the filament free-end describes a figure-eight shaped (or $\infty$-shaped) trajectory.

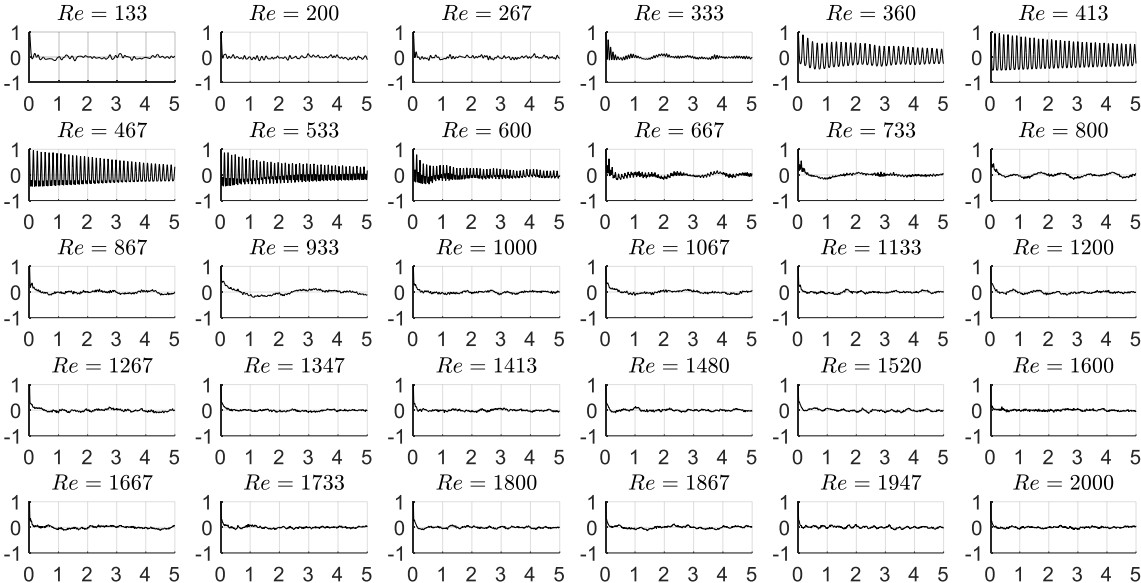

**Figure 8.** Autocorrelation function of the total displacement of the free-end of Filament 1 at different Reynolds number values (the *x*-axis scale is in seconds).

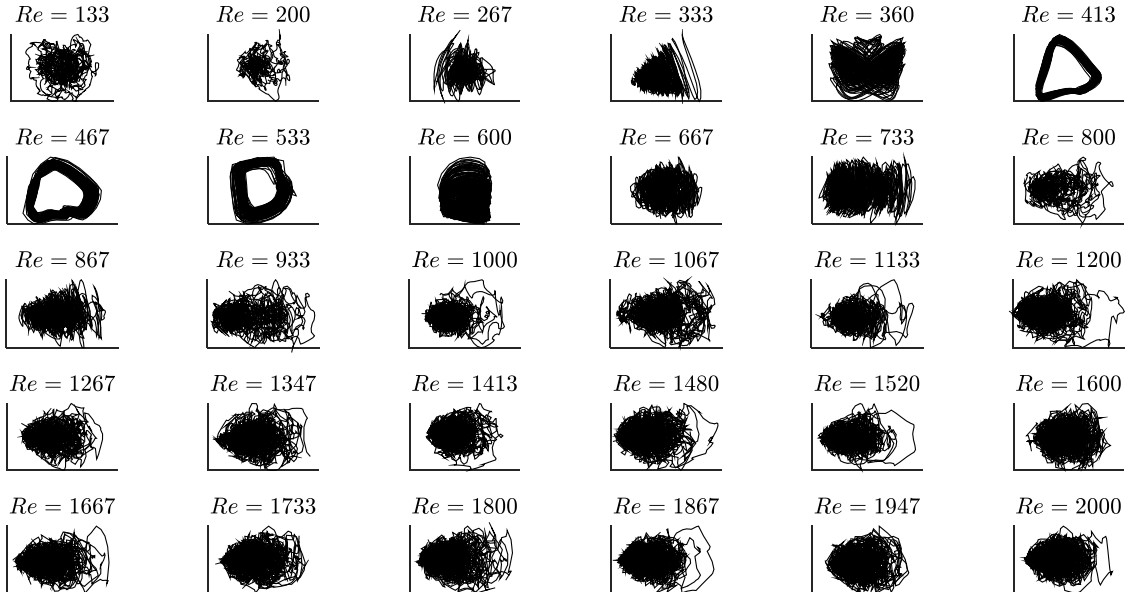

**Figure 9.** Reconstructed attractor in phase-space for the total displacement of the free-end of Filament 1 at different Reynolds number values.

Notably, the transition from the small-amplitude vibration to the large-amplitude limit-cycle oscillation is gradual and not abrupt: at $Re = 360$ the filament motion is large-amplitude, the autocorrelation function is already periodic and slowly decaying, though the attractor is still blob-like, thus indicating a dynamic intermediate between a small-amplitude vibration

and a large-amplitude limit-cycle oscillation. For Reynolds number beyond about $Re = 667$, the filament motion is large-amplitude, the autocorrelation function decays rapidly and the attractor becomes blob-like, indicating a large-amplitude non-periodic oscillation. Again, the transition from large-amplitude limit-cycle oscillation to large-amplitude non-periodic oscillation is gradual, as can be noticed at $Re = 600$ where the autocorrelation function is still periodic and rather slowly decaying, though the attractor is already blob-like. Finally, the dominant frequency of oscillation gradually increases with increasing Reynolds number.

Plots for Filaments 2, 3, 4, and 5 analogous to those for Filament 1 included in Figures 6–9 are provided in the Appendix A. As can be noted, the dynamics of Filaments 2–3 is qualitatively similar to that observed with Filament 1: as the flow velocity gradually increases, the structural response gradually evolves from a small-amplitude vibration to a large-amplitude limit cycle oscillation, and then into a large-amplitude non-periodic motion. With Filaments 4–5, on the other hand, the small-amplitude vibration evolves directly into a large-amplitude non-periodic motion. Whilst with Filament 4 there is a range of Reynolds number values where the large-amplitude oscillation becomes periodic, limit-cycle oscillations are no longer observed with Filament 5. In general, the Reynolds number values where the filament response changes gradually increase with decreasing filament length, indicating that higher flow velocities are required to trigger a change in structural response as the filament gradually shortens.

The Reynolds number range where limit-cycle oscillations are sustained widens when moving from Filament 1 (413–533) to Filament 2 (533–733) and then to Filament 3 (587-960), then contracts with Filament 4 (1053–1333). As previously noted, during limit-cycle oscillation with Filament 1 the filament free-end describes a figure-eight shaped trajectory. This is also the case with Filaments 2 and 3 but only for low Reynolds number values: for high Reynolds numbers the filament motion during limit-cycle oscillation tends to become two-dimensional and confined to the vertical plane. Notably, this is always the case with Filament 4, where the figure-eight shaped trajectory is not observed and limit-cycle oscillations are always two-dimensional.

Finally, the structural response of Filament 6 was reduced to a static deflection, i.e., the Filament 6 deflected, as the flow velocity was gradually increased, always maintaining a static equilibrium configuration. Static deflection angles of Filament 6 are provided in Figure 10 as function of the Reynolds number (the static deflection angle is defined as indicated in the insert in Figure 10).

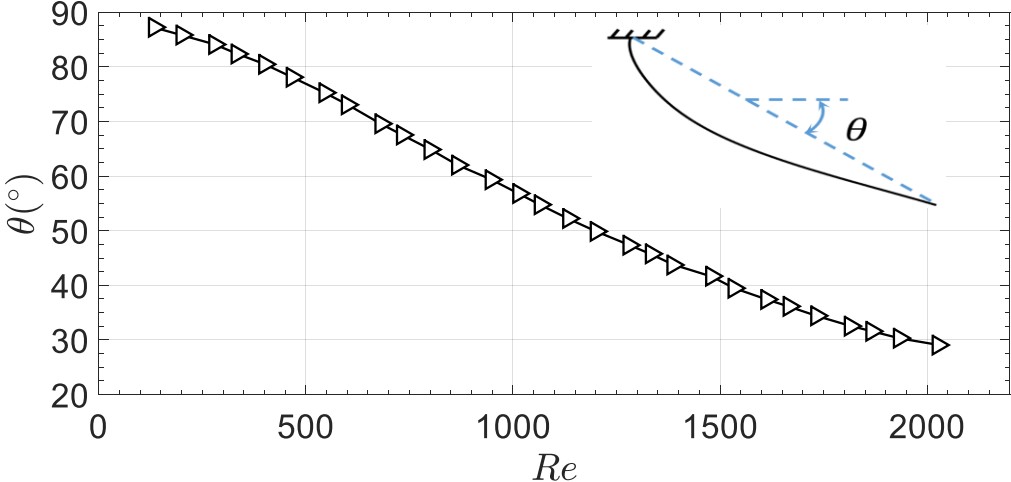

**Figure 10.** Static deflection angle $\theta$ for Filament 6 vs. Reynolds number.

As can be noted in Figure 10, the static deflection angle decreases with increasing Reynolds number following a sigmoidal trend, similarly to previous observations with circular cross-section filaments [42]. The sigmoidal trend of the static deflection angle indicates that the incremental change in inclination gradually decreases, as the flow velocity gradually increases. This trend can be explained by considering that, as the deflection gradually increases, the filament exposes a gradually smaller

frontal area to the flow, and consequently becomes gradually more streamlined. A qualitatively similar behavior has been observed with flexible plates [43,44] and flexible vegetation [45].

In order to better compare the structural responses of Filaments 1−5, the measurements are provided in aggregated form in Figure 11, where the displacements and frequencies are presented in dimensionless form as functions of the Reynolds number. The dimensionless displacement, in particular, is defined as follows:

$$A^* = \frac{A}{L} \tag{11}$$

where $A$ is the displacement (in either the horizontal or the vertical plane) and $L$ is the length of the filament (from Table 2). The dimensionless frequency, on the other hand, is defined as follows:

$$f^* = \frac{f}{f_1} \tag{12}$$

where $f$ is the flapping frequency (in either the horizontal or the vertical plane) and $f_1$ is the mode-1 fundamental frequency of vibration for each filament (from Table 2). As can be seen from the plot of the dimensionless displacement in the vertical direction ($A_y^*$ in Figure 11a), the onset of large-amplitude motion progressively occurs at higher Reynolds numbers as the length of the filament is decreased. The dimensionless displacement then increases as function of Reynolds number and levels off at a certain maximum. This maximum dimensionless displacement is progressively higher as the filament length is decreased. Moreover, it can be noted that, regardless of the Reynolds number for the onset of large-amplitude motion or the amplitude of motion, the dimensionless frequency in the vertical direction ($f_y^*$ in Figure 11c) exhibits a linearly increasing trend as function of the Reynolds number, indicating that at a higher wind speed corresponds a stronger fluid forcing and, therefore, a faster dynamic. As highlighted in Figure 11c, the onset of vibration in the vertical plane occurs at around $f_y/f_1 = 1$, therefore indicating that the filaments start vibrating (in the vertical plane) with a frequency that is comparable with their mode-1 natural vibration frequency. As can be seen from the plot of the dimensionless displacement in the horizontal direction ($A_z^*$ in Figure 11b), the Reynolds number ranges where the filaments motion tends to become two-dimensional and confined to the vertical plane are clearly recognizable. Other than this, the trends in Figure 11b are similar to those observed in Figure 11a, particularly so for Filament 1 whose dynamics is always three-dimensional. Dimensionless frequencies in the horizontal plane ($f_z^*$ in Figure 11d) grow approximately linearly with increasing Reynolds number, similarly to what observed in the vertical plane (Figure 11c). The spikes in dimensionless frequency observed in Figure 11d correspond to harmonics of the lowest peak frequency from the power spectral density, approximately corresponding to double of the lowest peak frequency. Similar to the case of vertical motion, the onset of large-amplitude motion for Filaments 1−3 occur at around $f_z/f_1 = 1$, indicating that these filaments start vibrating also in the horizontal plane with a frequency that is comparable with their mode-1 natural vibration frequency. Notably, for Filaments 4 and 5 the onset of motion is close to $f_z/f_1 = 2$, so that these filaments start vibrating in the horizontal plane with a frequency that is approximately twice their mode-1 natural vibration frequency.

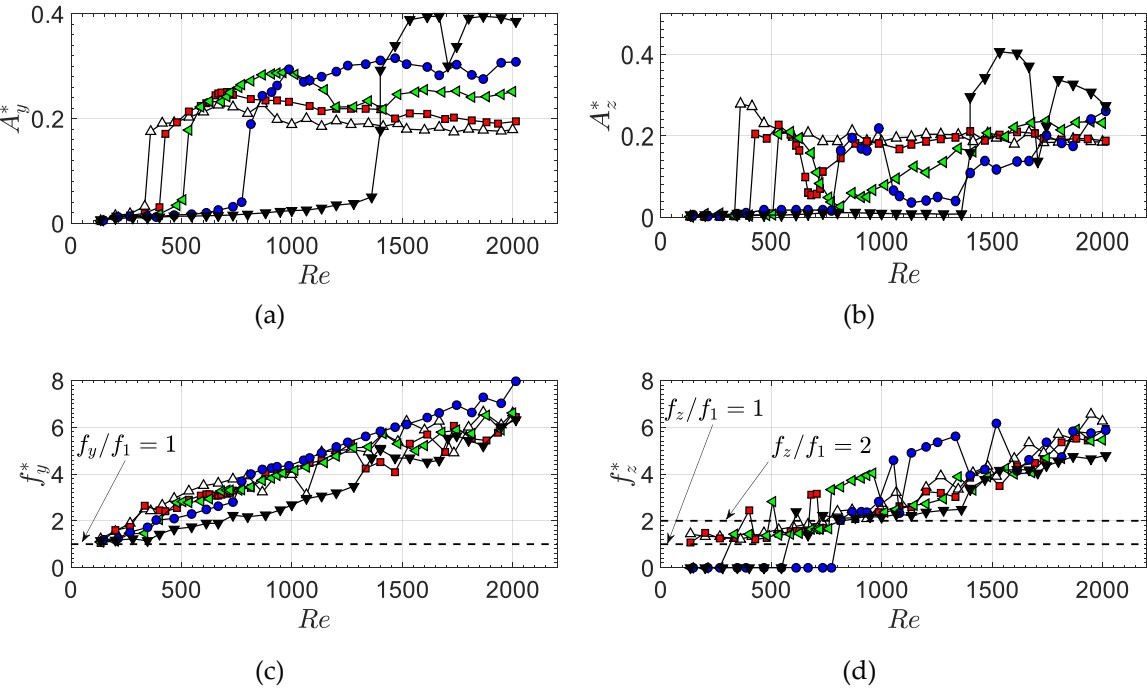

**Figure 11.** Dimensionless displacements and frequencies for Filaments 1−5 vs. Reynolds number. (**a**) Dimensionless displacement in the vertical plane; (**b**) Dimensionless displacement in the horizontal plane; (**c**) Dimensionless frequency in the vertical plane; (**d**) Dimensionless frequency in the horizontal plane. Legend: Filament 1 (white △); Filament 2 (red □); Filament 3 (green ◁); Filament 4 (blue ○); Filament 5 (black ▽).

A condensed representation of the observed filaments response is presented in the dynamics map provided in Figure 12a, where the observed filament dynamics is displayed as function of the filament length and Reynolds number, and in the stability map in Figure 12b, where the observed filament dynamics is displayed as function of the Scruton number *Sc* and reduced velocity *U**:

$$Sc = \frac{2m\zeta_1}{\rho L^2} \tag{13}$$

$$U^* = \frac{U}{f_1 L} \tag{14}$$

where $f_1$ and $\zeta_1$ are the first-mode natural vibration frequency and damping ratio of the filament (values provided in Table 2). Whilst the Scruton number can be regarded as a dimensionless representation of the filament damping, the reduced velocity be regarded as the ratio of the time-scale of the structural movement to the time-scale of the flow. As noted previously, the natural vibration frequency and the damping ratio of the filaments depend on the filament length. Accordingly, the filament length is used here as the representative linear dimension in place of the filament diameter, which is normally used with cross-flow-induced vibration. Even though the information conveyed by the dynamic map and the stability map in Figure 12 is essentially the same, the former is of more direct use for numerical methods validation, whereas the latter is more frequently used in the fluid-structure interaction literature.

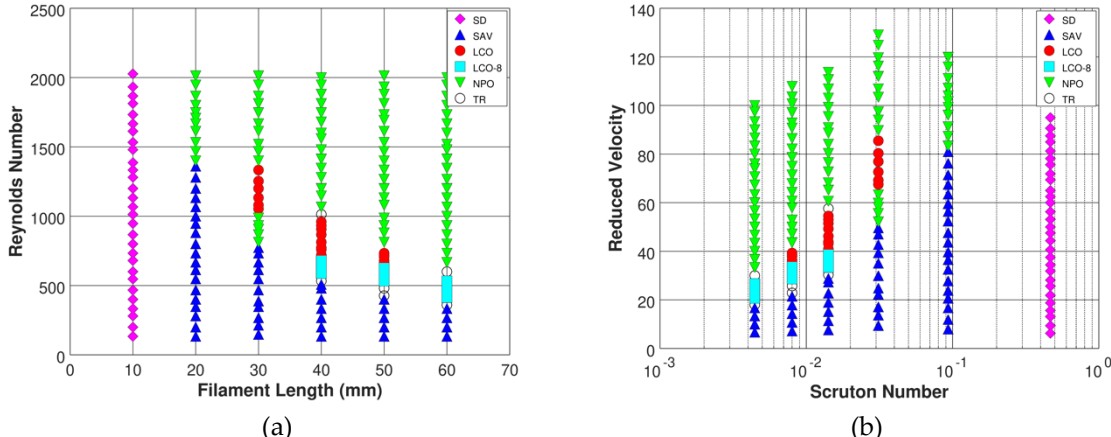

**Figure 12.** Condensed representation of the observed structural response of the flexible filaments. (**a**) Dynamic map; (**b**) Stability map. Legend: SD = static deflection; SAV = small-amplitude vibration; LCO = large-amplitude limit-cycle oscillation with two-dimensional trajectory; LCO-8 = large-amplitude limit-cycle oscillation with three-dimensional figure-eight-shaped trajectory; NPO = large-amplitude non-periodic oscillation; TR = transition.

It is evident that the observed filament responses are well separated and clustered in the dynamic and stability maps in Figure 12, thereby indicating that the structural response of the filaments is controlled by the Reynolds number (i.e., by the air flow velocity) and by the length of the filament or, equivalently, by the reduced velocity and Scruton number. The filament length clearly plays a central role in the structural response: as the filament length decreases the damping increases and so does the Scruton number, so that the excitation needed to trigger a transition or sustain a large-amplitude response increases, as it is evident in the corresponding increase of the Reynolds number and reduced velocity. For the shortest Filament 6, in particular, the damping is large enough to suppress any dynamic response within the flow velocity range explored, so that the structural response is reduced to a static deflection. The results highlight the importance of the filament damping ratio, which is modulated by the filament length, as a controlling parameter for the structural response. The importance of the filament length was already noted previously, when discussing the mode-1 natural vibration frequency and damping which also depend on the filament length. Finally, as it is evident from the reduced velocity values in Figure 12b (from about 5 up to about 130), the time-scale of the structure is much bigger than that of the flow, thereby indicating that the flow changes faster than the movement of the filaments. The interaction between the flow and the structure is not one-way, however, because the structural movement is large enough to significantly modify the flow field. The present results are in qualitative agreement with documented observations of flexible filaments of circular cross-section in air flow [32,33]. A notable difference is that the large-amplitude limit-cycle oscillation where the filament free-end describes a figure-eight-shaped trajectory documented here was not observed with circular cross-section filaments, which suggests that reducing the symmetry of the filament cross-section may yield a richer dynamic.

## 4. Conclusions

We presented results of an experiment specifically designed for the validation of numerical methods for aeroelasticity and FSI problems, and intended to complement and extend available benchmark validation test cases. The experiments were conducted in a wind tunnel, using flexible filaments of rectangular cross-section and varying length whose dynamics was recorded via fast-video imaging. The Reynolds number range covered corresponds to laminar and mildly turbulent flow conditions. The structural response of the filaments is modulated by the Reynolds number (i.e., by the air flow velocity) and by the filament length, and includes: (1) static reconfiguration, (2) small-amplitude

vibration, (3) large-amplitude limit-cycle periodic oscillation, and (4) large-amplitude non-periodic motion. The damping of the flexible filaments, which is controlled by the filament length, plays a central role in the structural response. The experimental results presented herein are valuable for the validation of numerical methods for aeroelasticity and, more generally, for fluid-structure interaction applications.

**Supplementary Materials:** The following are available online at http://www.mdpi.com/2311-5521/5/2/90/s1, Raw wind speed time series (unit m/s) from flow characterization stored in comma-separated value file 'free_stream_raw_data.dat'. Each column of csv file 'free_stream_raw_data.dat' corresponds to a fixed wind speed setting (in m/s). Sampling time: 5 seconds; sampling frequency: 20 kHz.

**Author Contributions:** Conceptualization, J.S.-L. and A.C.; experiments, J.S.-L.; formal analysis, J.S.-L. and A.C.; writing—original draft preparation, J.S.-L. and A.C.; writing—review and editing, J.S.-L. and A.C. All authors have read and agreed to the published version of the manuscript.

**Funding:** This research received no external funding.

**Acknowledgments:** Andrew Kennaugh from the Department of Mechanical, Aerospace and Civil Engineering of the University of Manchester (UK) is gratefully acknowledged for his technical support.

**Conflicts of Interest:** The authors declare no conflict of interest.

## Appendix A

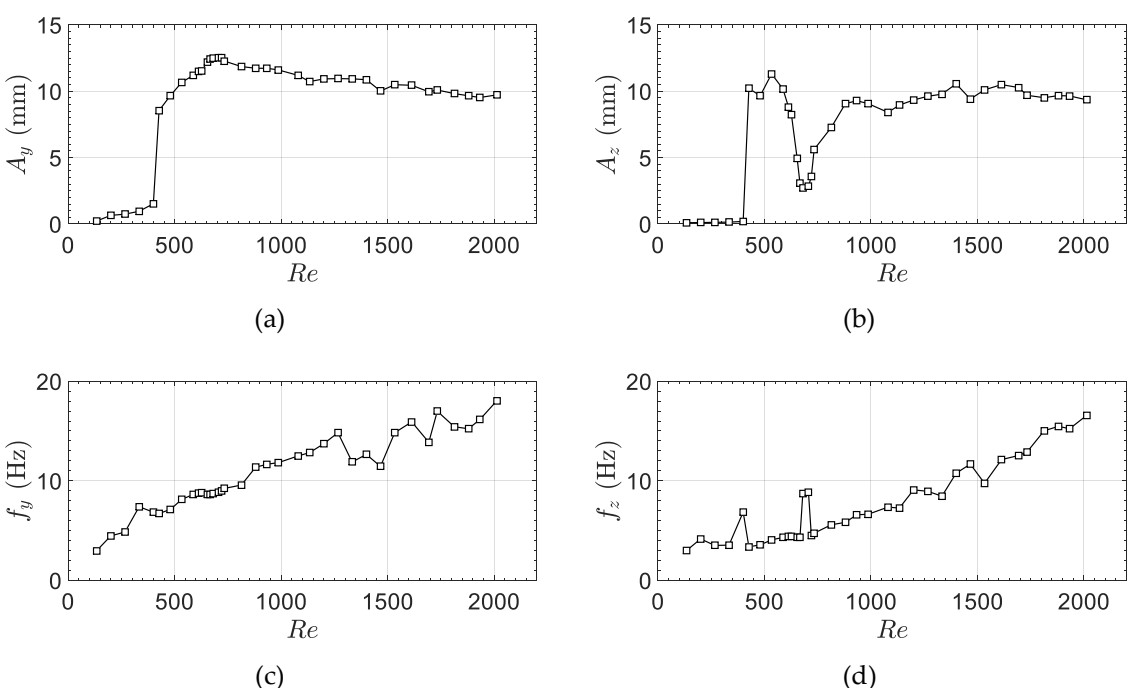

(a)

(b)

(c)

(d)

**Figure A1.** Displacements and flapping frequencies measured for Filament 2. (**a**) Displacement in the vertical plane; (**b**) Displacement in the horizontal plane; (**c**) Frequency in the vertical plane; (**d**) Frequency in the horizontal plane.

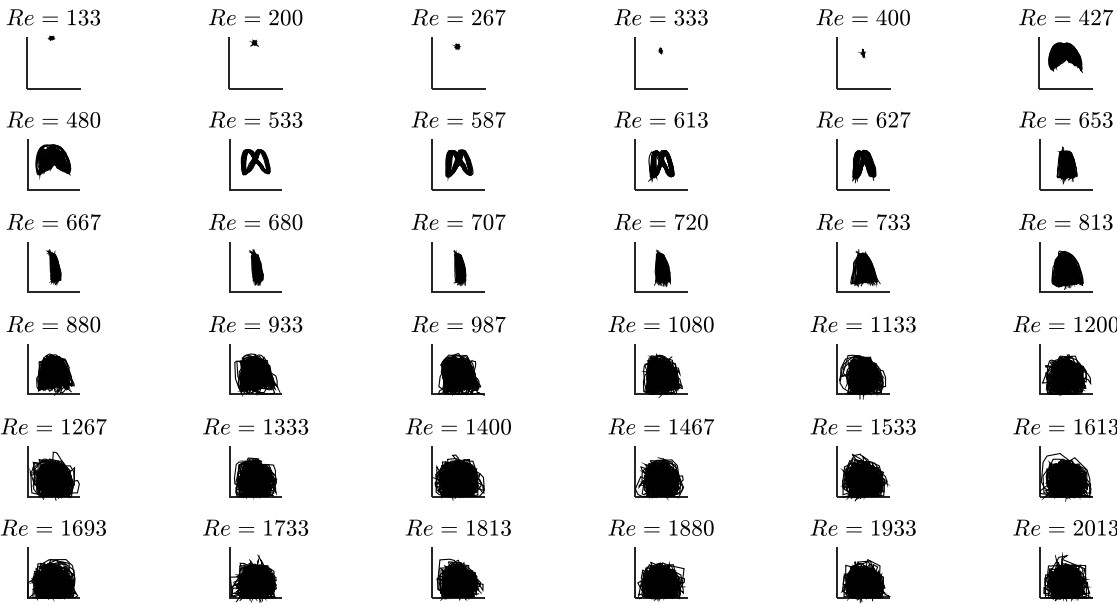

**Figure A2.** Trajectory of the free-end of Filament 2 as seen by an observer located downstream of the filament and facing the flow (Y-Z plane in Figure 2c).

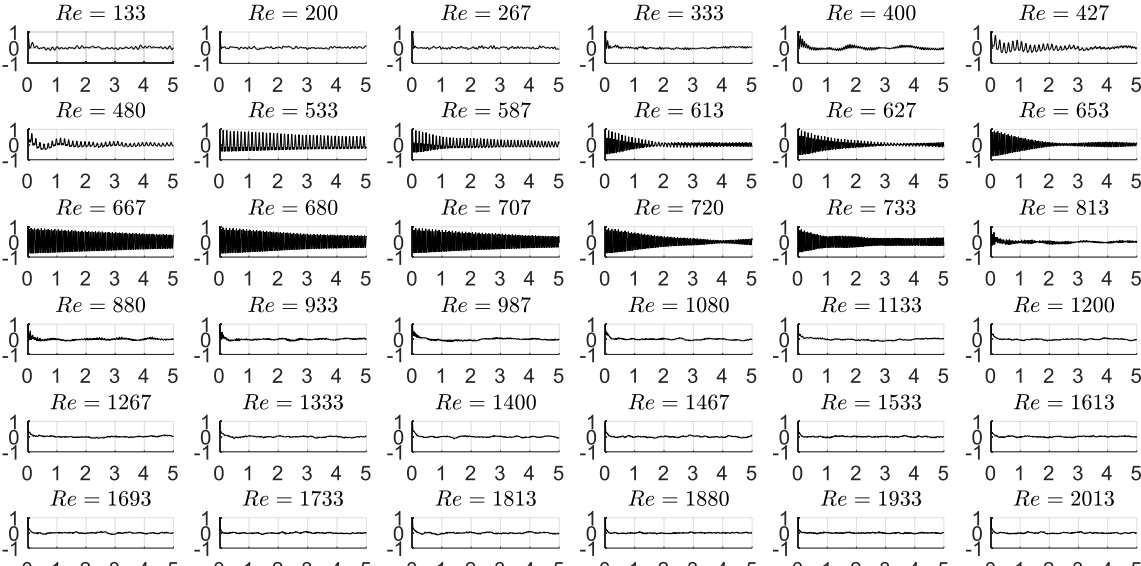

**Figure A3.** Autocorrelation function of the total displacement of the free-end of Filament 2 at different Reynolds number values (the *x*-axis scale is in seconds).

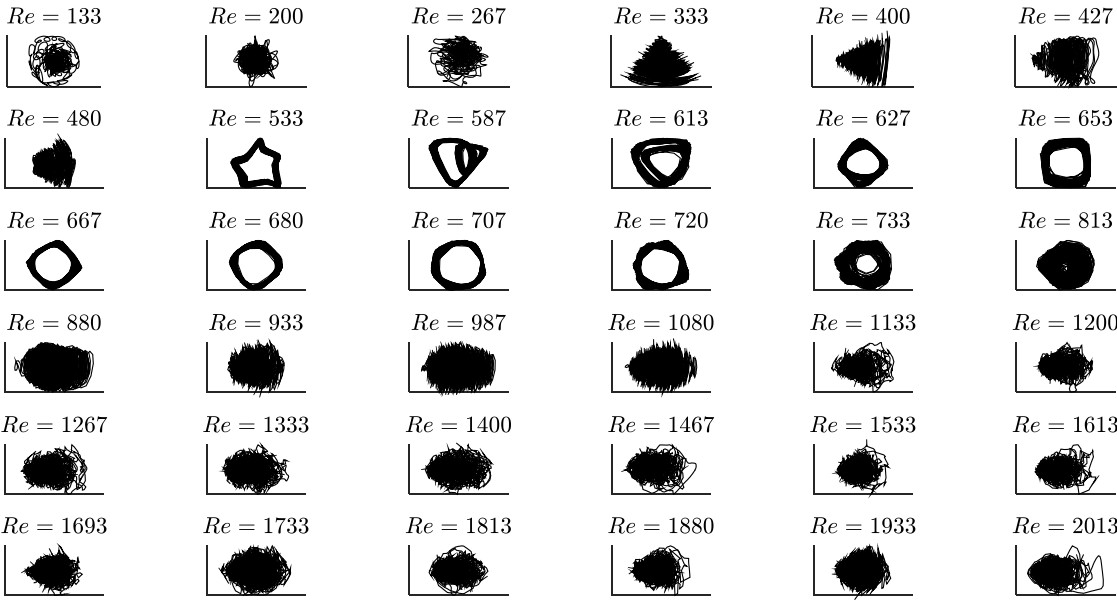

**Figure A4.** Reconstructed attractor in phase-space for the total displacement of the free-end of Filament 2 at different Reynolds number values.

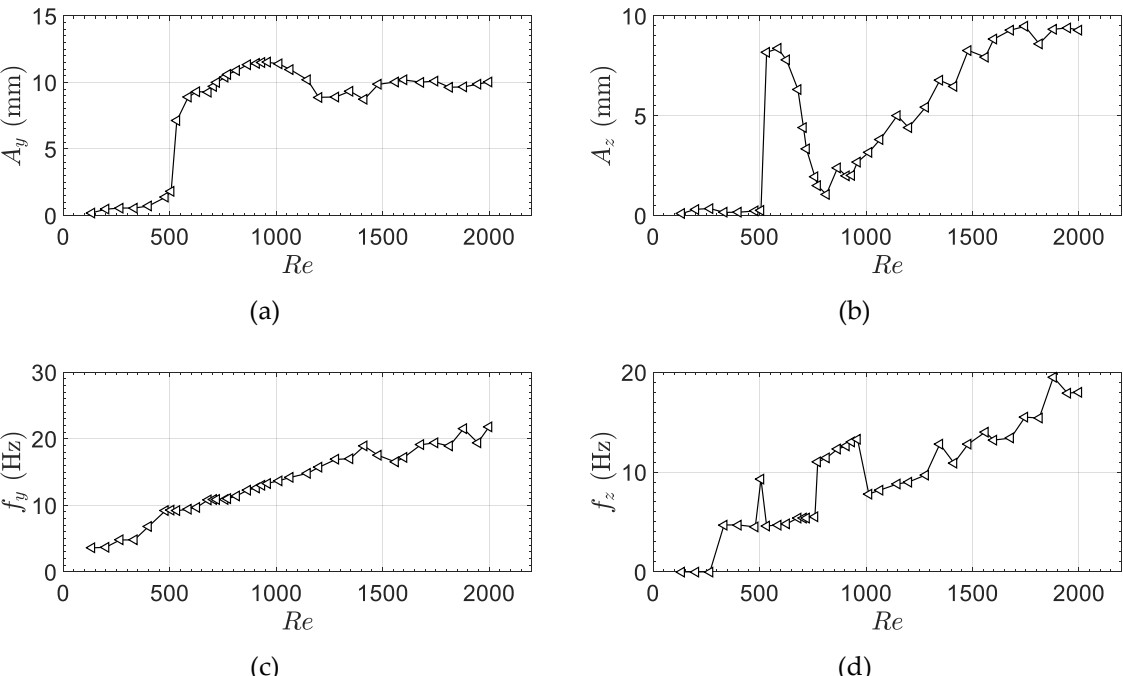

**Figure A5.** Displacements and flapping frequencies measured for Filament 3. (**a**) Displacement in the vertical plane; (**b**) Displacement in the horizontal plane; (**c**) Frequency in the vertical plane; (**d**) Frequency in the horizontal plane.

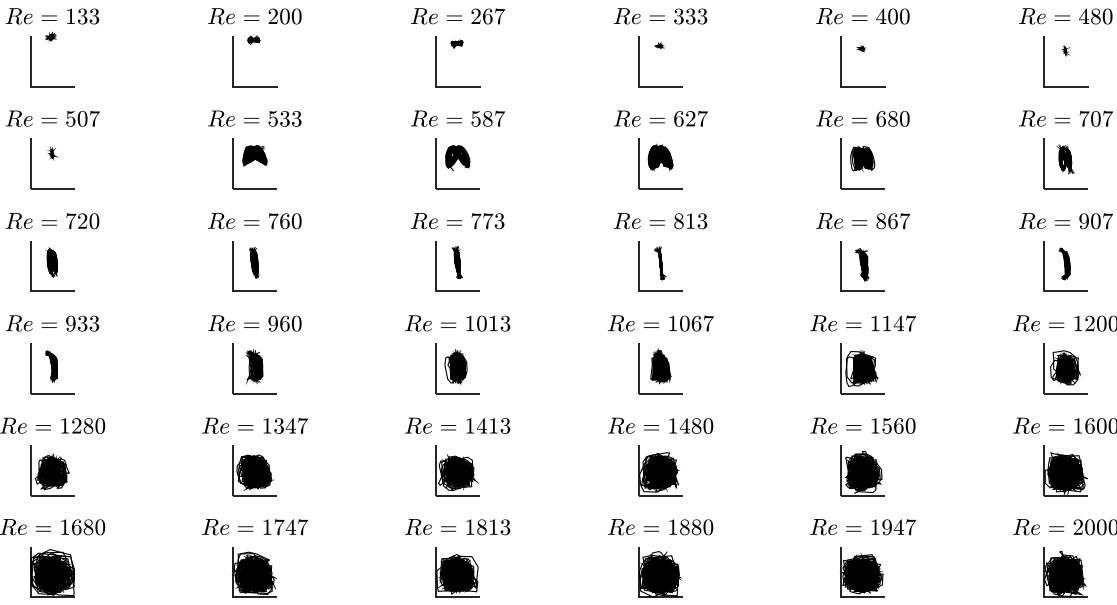

**Figure A6.** Trajectory of the free-end of Filament 3 as seen by an observer located downstream of the filament and facing the flow (Y-Z plane in Figure 2c).

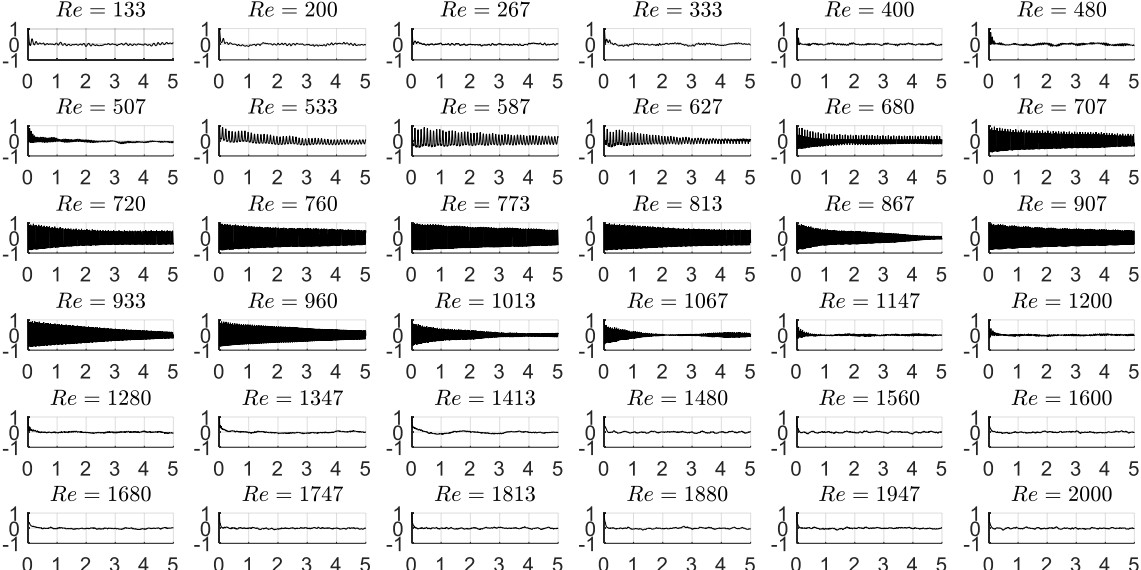

**Figure A7.** Autocorrelation function of the total displacement of the free-end of Filament 3 at different Reynolds number values (the *x*-axis scale is in seconds).

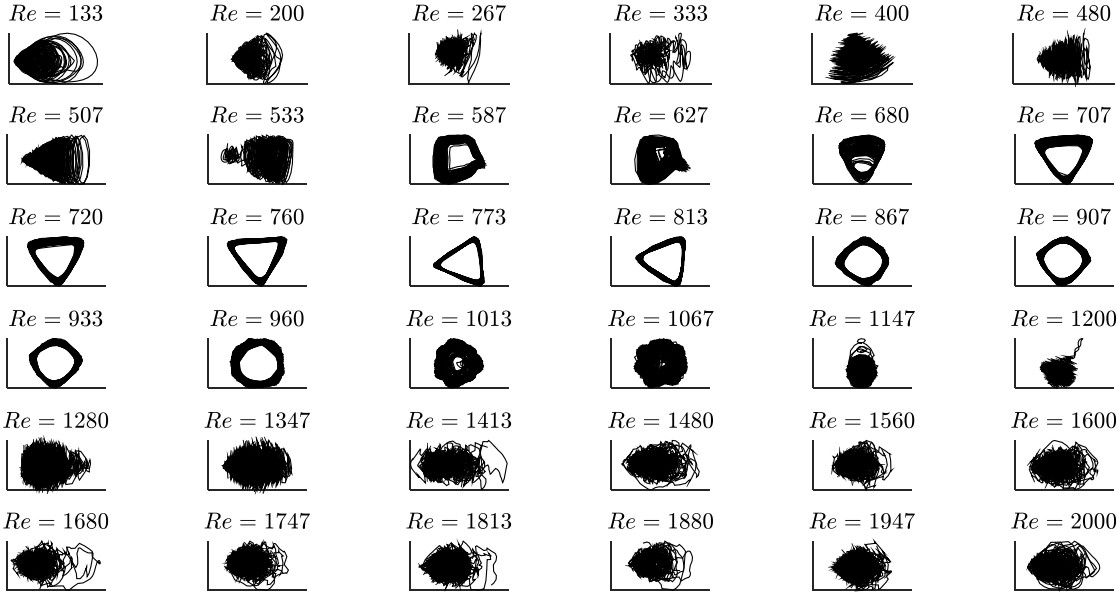

**Figure A8.** Reconstructed attractor in phase-space for the total displacement of the free-end of Filament 3 at different Reynolds number values.

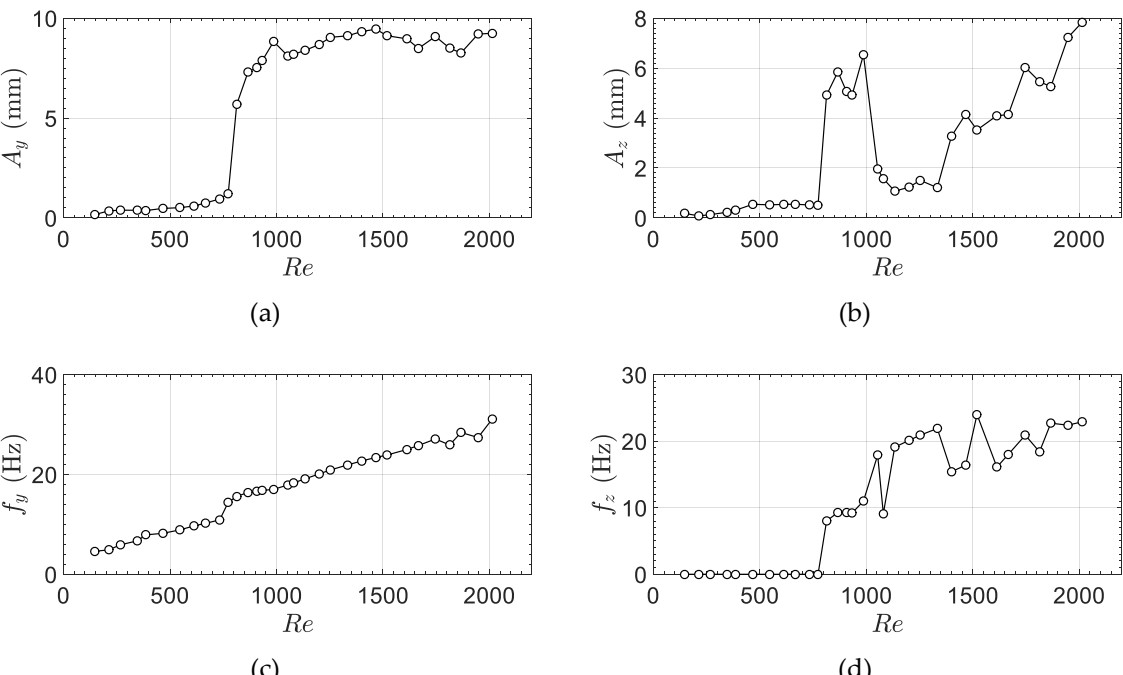

**Figure A9.** Displacements and flapping frequencies measured for Filament 4. (**a**) Displacement in the vertical plane; (**b**) Displacement in the horizontal plane; (**c**) Frequency in the vertical plane; (**d**) Frequency in the horizontal plane.

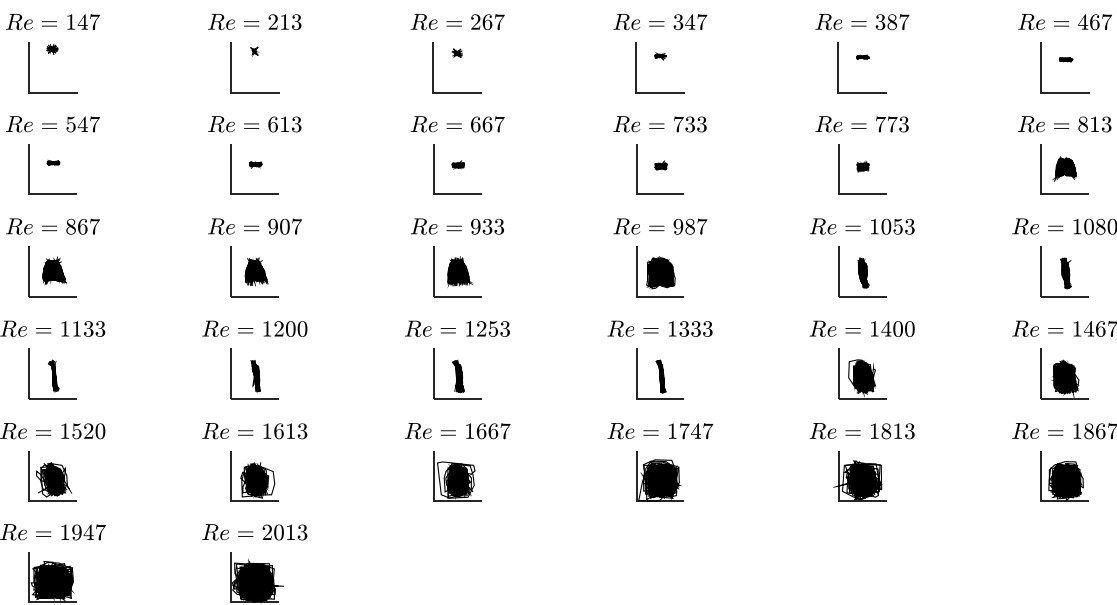

**Figure A10.** Trajectory of the free-end of Filament 4 as seen by an observer located downstream of the filament and facing the flow (Y-Z plane in Figure 2c).

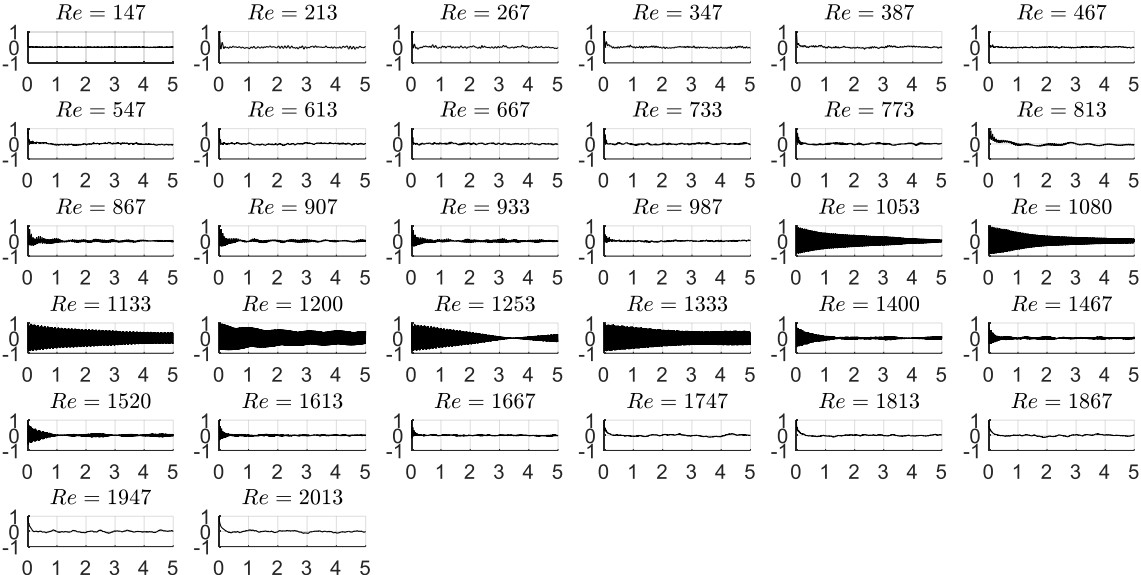

**Figure A11.** Autocorrelation function of the total displacement of the free-end of Filament 4 at different Reynolds number values (the *x*-axis scale is in seconds).

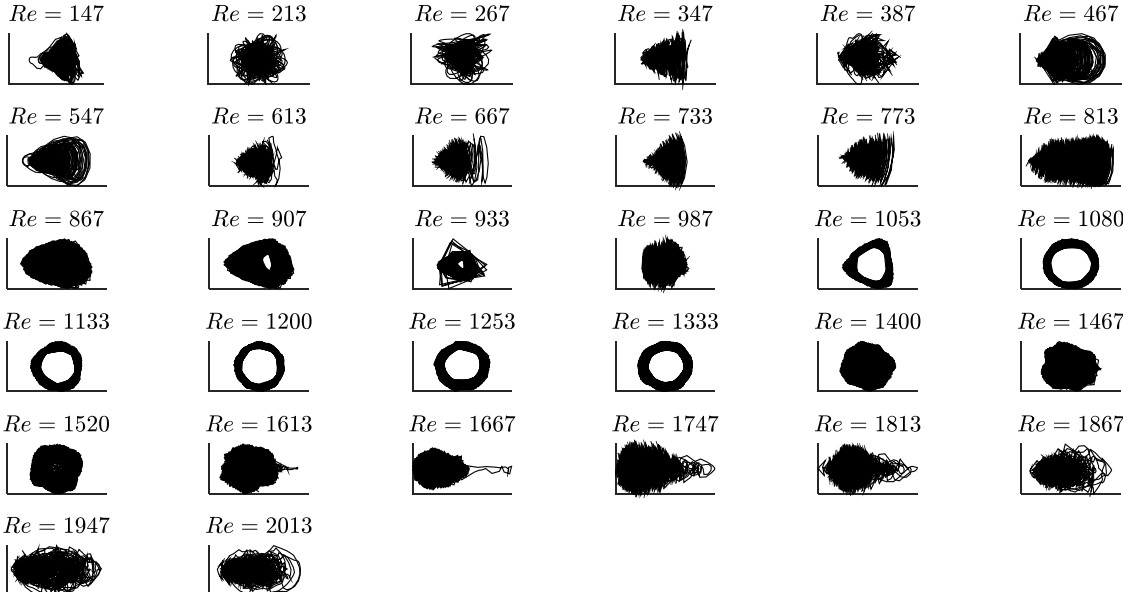

**Figure A12.** Reconstructed attractor in phase-space for the total displacement of the free-end of Filament 4 at different Reynolds number values.

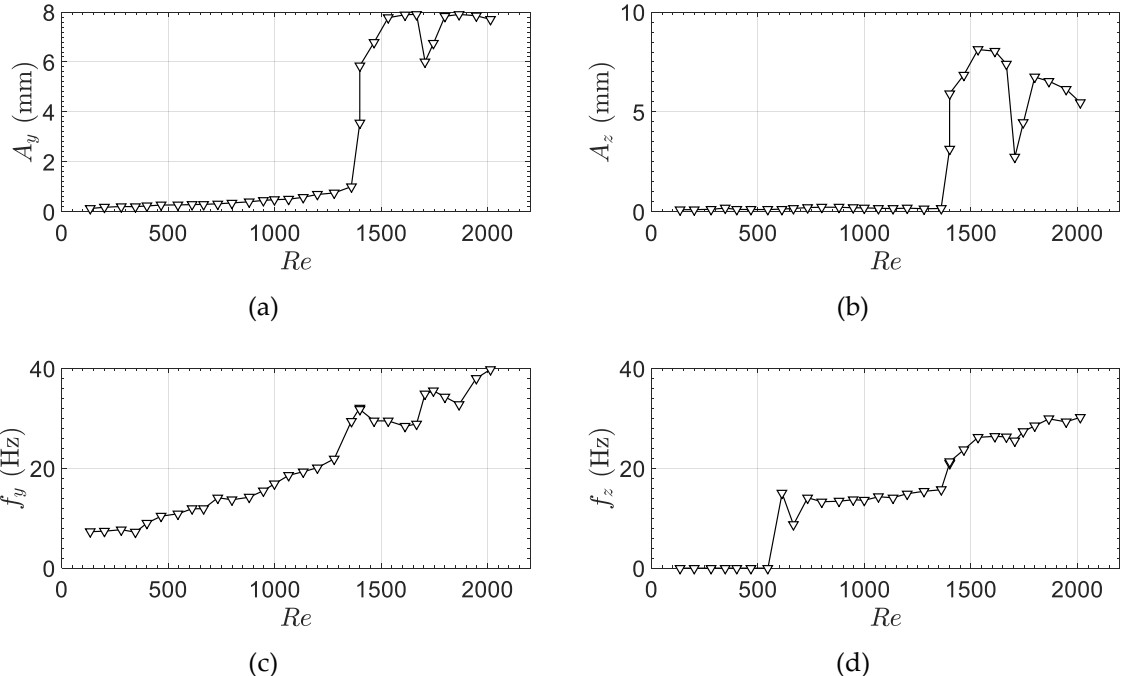

**Figure A13.** Displacements and flapping frequencies measured for Filament 5. (**a**) Displacement in the vertical plane; (**b**) Displacement in the horizontal plane; (**c**) Frequency in the vertical plane; (**d**) Frequency in the horizontal plane.

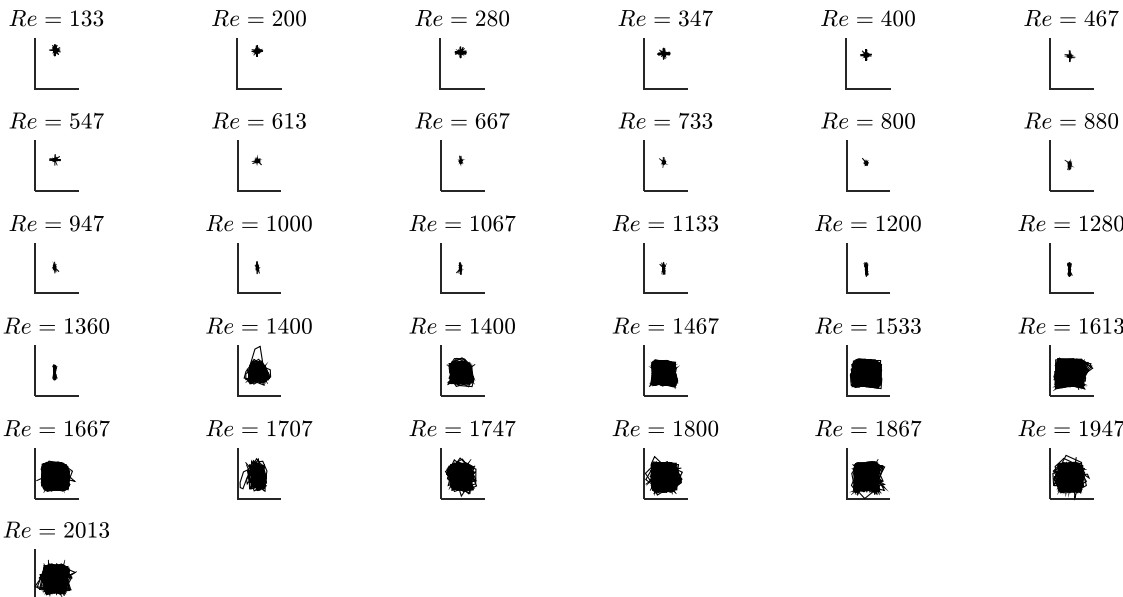

**Figure A14.** Trajectory of the free-end of Filament 5 as seen by an observer located downstream of the filament and facing the flow (Y-Z plane in Figure 2c).

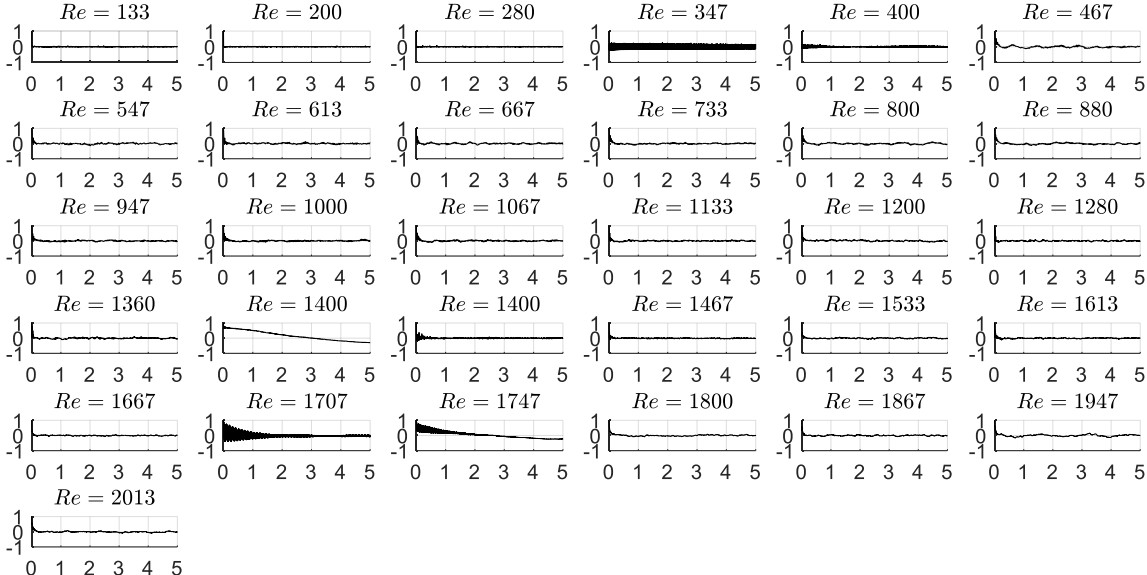

**Figure A15.** Autocorrelation function of the total displacement of the free-end of Filament 5 at different Reynolds number values (the *x*-axis scale is in seconds).

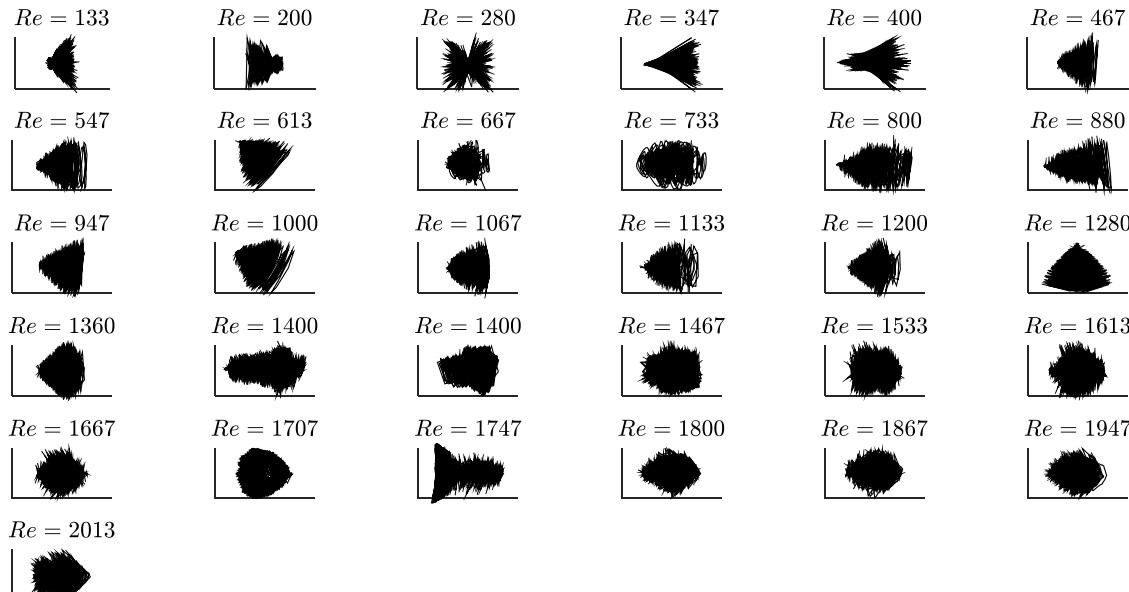

**Figure A16.** Reconstructed attractor in phase-space for the total displacement of the free-end of Filament 5 at different Reynolds number values.

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
