# Peer review of "Experiments on Flexible Filaments in Air Flow for Aeroelasticity and Fluid-Structure Interaction Models Validation"

_fluids, doi:10.3390/fluids5020090_

Round 1

Reviewer 1 Report

Dear authors,

Please find attached my comments on your paper.

Your work is very interesting. So I think its presentation should be improved for more consistency and interest for the readers.

Regards

Author Response

In this paper, the authors present results of an experimental study conducted in a wind tunnel. Six flexible filaments of rectangular cross-section and varying length, which were vertically oriented in cantilever-beam configuration and exposed to air cross-flow of moderate Reynolds number, corresponding to laminar and mildly turbulent flow conditions, are tested in order to explore a wider range of structural responses. It is observed that the dynamics of the flexible filaments was generally 3D, though a two-dimensional structural response was sometimes observed during limit-cycle periodic oscillations. In addition, the structural response of the filaments included static reconfiguration, small-amplitude vibration, large-amplitude limit-cycle periodic oscillation and large- amplitude non-periodic motion. The purpose of the study is to provide experimental results valuable for the validation of numerical FSI methods at large, without focusing on any particular application. While this is definitely an area worthy of investigation, there are some remarks about the content and the results presented in the paper as detailed below.

1) P.6, 2nd paragraph (after eq. 5), line 5:  the filaments 1 and 2 are presented as the longer filaments in table 2 and the filaments 3 through 6 are the shorter. That’s not the case here. Please make the corrections.

This has been corrected.

2) P.7, line 7: … can be considered as a measure of …

This has been corrected as suggested.

3) In the figures “Displacements and flapping frequencies …” (6, A1, A5, A9 and A13) the graphics (b) represent the displacement and (c) the frequency. Please reverse.

 This has been corrected.

4) Why does the uncertainty on the linear density (P.3) seem too high (around 20%), while it is around 5% on the other quantities (density, length, etc.)?  

To avoid the large error that would have arisen from measuring directly the mass of the filament, which was on the order of 0.1g, the density was deduced by measuring the mass of a bigger chunk of silicon rubber (with a mass on the order of a few grams). The linear mass density is therefore calculated as the product of the silicon rubber density (measured with ±5% error) times the height (measured with ±12.5% error) and width (measured with ±2.5% error) of the filament cross section. The error in the linear mass density is therefore of around 20% (5% + 2.5% + 12.5%). The following paragraph is now included in the revised manuscript to better explain this point:

Note that, in order to avoid the large error that would have arisen from measuring directly the mass of the filament (which was on the order of 0.1 g), the density of the silicon rubber provided above was deduced from measuring the mass of a bigger chunk of silicon rubber. The linear mass density of the filament provided above was therefore calculated as the product of the silicon rubber density times the width and height of the filament cross-section.

5) On P.5, a forced vibration test under single frequency excitation is mentioned. How this test was performed and what is the frequency was used? How the natural frequency of the filament was measured in this case? Is there a relationship between these two frequencies?

The following paragraph is now included in the revised manuscript to better explain this point:

The test setup included an electromagnetic shaker (of in-house design and construction) with control signal provided by a signal generator operated in sine wave mode with frequency resolution of 0.1 Hz. During the tests, the filaments hang vertically with the top extreme fixed to the shaker. Following common practice, the amplitude of response of the filament was recorded (using a Panasonic Lumix DMC-FZ200 digital camera) as a function of the excitation frequency, and the natural vibration frequency was identified as the peak in the response (experimental uncertainty deduced from the full-width at half maximum of the peak in the amplitude response).

6) How the damping test was performed and how the damping coefficient of the filament was measured?

The following paragraph is now included in the revised manuscript to better explain this point:

On the other hand, first-mode damping ratios  were deduced from free vibration tests in stagnant air. Starting with the filament hanging vertically in equilibrium with the top extreme fixed, the filament free-end was manually displaced (displacement small enough to trigger a mode-1 response). The filament free-end was then released, and the free vibration of the filament was recorded (using a Panasonic Lumix DMC-FZ200 digital camera). Following common practice, the damping ratio was finally evaluated from the logarithmic decrement of the envelope of the displacement time-series (experimental uncertainty deduced as standard deviation from repeated measurements)

7) Where is located the laminar flow regime in the figure 4 (a)?

Below about 1.5 m/s, based on the Reynolds number in Eq. (9). This is now explained in the revised manuscript as follows:

The Reynolds number range explored here, therefore, covers laminar (flow velocity up to ~1.5 m/s), transitional (flow velocity from ~1.5 m/s up to ~2.5 m/s), and mildly turbulent (flow velocity above ~2.5 m/s) flow conditions.

8) I think that the validation of the numerical FSI models cannot be dissociated from the application field concerned. For example, models with large deformations are not the same as models with small deformations. 3D models are not the same as 2D models. The models with complex flows (high turbulence, multiphasic, cavitation, etc.) are not the same as the models with more standard flows. The present study concerns the 3D large deformations structures, exposed to a cross air flow of various moderate Reynolds numbers. Hence, the results presented here can be used for numerical models validation only for this case. I suggest that the paper be focused on the validation of FSI numerical models related on these application cases and not on the validation of all FSI numerical models in general. This is already partially the case in Table 1, where the bibliographic list does not representative of the literature in this field.

In response to the reviewer’s comment, the abstract and introduction have been edited and the focus of the work realigned towards aeroelasticity applications. The title has been updated accordingly:

Experiments on Flexible Filaments in Air Flow for Aeroelasticity and Fluid-Structure Interaction Models Validation

We however believe that the present results can be of interest also beyond aeroelasticity applications, even for FSI models not intended for aeroelastic applications. It is not infrequent, in fact, that numerical models are validated against several experimental benchmarks including some outside the range of immediate interest, even only to check the asymptotic consistency or robustness of the code outside the intended field of application. The following paragraph is now included in the revised manuscript to better explain this point:

Even though the focus here is clearly on three-dimensional large deformations of flexible structures in air flow, the practical relevance of the work goes beyond aeroelasticity applications, and the present results can be of interest for FSI applications in general.

9) The physical analysis in FSI problems concerns the dynamics of both structures and fluids. Therefore, the validation of numerical models must also cover the both aspects. Which is not the case in this study. Indeed, only the results on the dynamics of the structure are presented. So I wonder what justifies the title and the orientation of this paper to the FSI. I think that it is more useful for the reader to present this work as a numerical models validation test case for a deformable structure response to a cross flow, without mentioning the FSI.

The problem investigated is a FSI problem, and in our opinion the fact that the flow field is not measured does not change the nature of the problem. The title and orientation of the paper have been realigned (see reply to the previous point), but we prefer to keep the FSI terminology. As already noted in the original manuscript, measuring the flow field was not feasible in the present case. Having a flexible structure of small size allows observing large deformations with small Reynolds number, which is presently preferred with FSI validation test cases. The side effect of having a small structure is that the details of the flow also become small, making the visualization of the flow field challenging or impractical (in general, not only for the PIV setup that we had available). This is now better explained in the revised manuscript as follows:

As previously noted, in FSI test cases it is normally preferred to have flexible structures which undergo large deformations with moderate motion frequency whilst interacting with a flow of moderate Reynolds number, thereby avoiding the complications of simulating highly turbulent flows. The small size and high flexibility of the present filaments were instrumental in achieving large displacements and relatively small oscillation frequencies, and at the same time keep the Reynolds number small. Unfortunately, the details of the flow field scale with the size of the structure, so that the smaller the structure the smaller the space resolution that is needed to faithfully resolve the flow field. In the present case, the requirements for a faithful flow field visualization were beyond our experimental capabilities, and the flow field was therefore not measured.

10) In general, the physical analysis of the results of this work must be improved. Very little physical interpretation of the curves and figures is given by the authors, while many results have been produced. In addition, a detailed comparative study of the different cases studied is desired in order to better enlighten the reader.

The physical analysis has been significantly expanded, and a comparative study of the different cases included. The following text is now included in the revised manuscript:

In order to better compare the structural responses of Filaments 1-5, the measurements are provided in aggregated form in Figure 11, where the displacements and frequencies are presented in dimensionless form as functions of the Reynolds number. The dimensionless displacement, in particular, is defined as follows:

[see Eq. 11]

where  is the displacement (in either the horizontal or the vertical plane) and  is the length of the filament (from Table 2). The dimensionless frequency, on the other hand, is defined as follows:

[see Eq. 12]

where  is the flapping frequency (in either the horizontal or the vertical plane) and  is the mode-1 fundamental frequency of vibration for each filament (from Table 2). As can be seen from the plot of the dimensionless displacement in the vertical direction (in Figure 11a), the onset of large-amplitude motion progressively occurs at higher Reynolds numbers as the length of the filament is decreased. The dimensionless displacement then increases as function of Reynolds number and levels off at a certain maximum. This maximum dimensionless displacement is progressively higher as the filament length is decreased. Moreover, it can be noted that, regardless of the Reynolds number for the onset of large-amplitude motion or the amplitude of motion, the dimensionless frequency in the vertical direction (in Figure 11c) exhibits a linearly increasing trend as function of the Reynolds number, indicating that at a higher wind speed corresponds a stronger fluid forcing and, therefore, a faster dynamic. As highlighted in Figure 11, the onset of vibration in the vertical plane occurs at around, therefore indicating that the filaments start vibrating (in the vertical plane) with a frequency that is comparable with their mode-1 natural vibration frequency. As can be seen from the plot of the dimensionless displacement in the horizontal direction ( in Figure 11b), the Reynolds number ranges where the filaments motion tends to become two-dimensional and confined to the vertical plane are clearly recognizable. Other than this, the trends in Figure 11b are similar to those observed in Figure 11a, particularly so for Filament 1 whose dynamics is always three-dimensional. Dimensionless frequencies in the horizontal plane ( in Figure 11d) grow approximately linearly with increasing Reynolds number, similarly to what observed in the vertical plane (Figure 11c). The spikes in dimensionless frequency observed in Figure 11d correspond to harmonics of the lowest peak frequency from the power spectral density, approximately corresponding to double of the lowest peak frequency. Similar to the case of vertical motion, the onset of large-amplitude motion for Filaments 1-3 occur at around, indicating that these filaments start vibrating also in the horizontal plane with a frequency that is comparable with their mode-1 natural vibration frequency. Notably, for Filaments 4 and 5 the onset of motion is close to, so that these filaments start vibrating in the horizontal plane with a. frequency that is approximately twice their mode-1 natural vibration frequency.

It is evident that the observed filament responses are well separated and clustered in the dynamic and stability maps in Figure 12, thereby indicating that the structural response of the filaments is controlled by the Reynolds number (i.e. by the air flow velocity) and by the length of the filament or, equivalently, by the reduced velocity and Scruton number. The filament length clearly plays a central role in the structural response: as the filament length decreases the damping increases and so does the Scruton number, so that the excitation needed to trigger a transition or sustain a large-amplitude response increases, as it is evident in the corresponding increase of the Reynolds number and reduced velocity. For the shortest Filament 6, in particular, the damping is large enough to suppress any dynamic response within the flow velocity range explored, so that the structural response is reduced to a static deflection. The results highlight the importance of the filament damping ratio, which is modulated by the filament length, as a controlling parameter for the structural response. The importance of the filament length was already noted previously, when discussing the mode-1 natural vibration frequency and damping which also depend on the filament length. Finally, as it is evident from the reduced velocity values in Figure 12b (from about 5 up to about 130), the time scale of the structure is much bigger than that of the flow, thereby indicating that the flow changes faster that the movement of the filaments. The interaction between the flow and the structure is not one-way, however, because the structural movement is large enough to significantly modify the flow field. The present results are in qualitative agreement with documented observations of flexible filaments of circular cross-section in air flow [32,33]. A notable difference is that the large-amplitude limit-cycle oscillation where the filament free-end describes a figure-eight-shaped trajectory documented here was not observed with circular cross-section filaments, which suggests that reducing the symmetry of the filament cross-section may yield a richer dynamic.

Reviewer 2 Report

  1. In the abstract, you mentioned that the problem of flow induced vibration can not be solved analytically, however, there is several fluidelastic instability models in the literature addressing the mechanism responsible to induce instability in the system. 
  2. Also, you talked about the validation of numerical simulations but you didn't simulations for the tests. Could you clarify that?
  3. In section 2.2, Why did you measure the natural frequency using forced vibration shaker? It can be estimated directly from free vibration test. What is the type of shaker, is it mechanical or electromagnetic shaker? In case of mechanical shaker, how did you mitigate the interactions that could occurred between the shaker and filament?
  4. What are the flapping frequencies of filaments relative to the natural frequency of the filament?
  5. Why the dominant frequency of oscillation increased with increasing Reynolds number? 
  6. The conclusions about the structural response variation with flow velocity are basically observed in the literature (see Paidoussis' work). Could you add another findings from your experiments?
  7. In page 13, you mentioned that  the filament dynamics is correlated with Scruton number and reduced velocity. This is correct for the response due to vortex induced vibrations, however at unstable limit cycle oscillations, the excitation mechanism is totally different, it would be fluidelastic instability. Could you explain?
  8. What is the added contribution of this paper relative to your published papers (1.Flow-Induced Motions of Flexible Filaments Hanging in Cross-Flow/2.Modulation of flexible filaments dynamics due to attachment angle relative to the flow)

Author Response

1) In the abstract, you mentioned that the problem of flow induced vibration cannot be solved analytically, however, there is several fluidelastic instability models in the literature addressing the mechanism responsible to induce instability in the system. 

In the abstract we refer broadly to FSI problems (which of course include FIV) and say that these problems cannot ‘normally’ be solved analytically, which does not imply that analytical solutions cannot be obtained in selected, simplified cases (such as the fluidelastic instability FIV models referred to by the Reviewer).

 2) Also, you talked about the validation of numerical simulations but you didn't simulations for the tests. Could you clarify that?

The objective of this work, which is purely experimental, is to provide experimental data that others will use to validate their numerical simulations.  This is stated clearly throughout the manuscript.

3) In section 2.2, Why did you measure the natural frequency using forced vibration shaker? It can be estimated directly from free vibration test. What is the type of shaker, is it mechanical or electromagnetic shaker? In case of mechanical shaker, how did you mitigate the interactions that could occurred between the shaker and filament?

Yes, the natural frequency can be deduced from free-vibration experiments (mode-1, for higher modes it becomes more challenging). Equivalently, the natural frequency can be measured in shaker tests; so that picking one technique or another is a matter of personal choice/convenience, as the final results are the same. Since we have used a shaker in our previous experiments with flexible filaments of circular cross-section (already published), for consistency we preferred to use the same technique also in these tests. The following paragraph is now included in the revised manuscript to better explain this point:

The test setup included an electromagnetic shaker (of in-house design and construction) with control signal provided by a signal generator operated in sine wave mode with frequency resolution of 0.1 Hz. During the tests, the filaments hang vertically with the top extreme fixed to the shaker. Following common practice, the amplitude of response of the filament was recorded (using a Panasonic Lumix DMC-FZ200 digital camera) as a function of the excitation frequency, and the natural vibration frequency was identified as the peak in the response (experimental uncertainty deduced from the full-width at half maximum of the peak in the amplitude response).

 4) What are the flapping frequencies of filaments relative to the natural frequency of the filament? In the revised manuscript, we have included a section on the comparative analysis of the response of different filament which also includes a plot of the flapping frequencies normalized to the natural frequencies. As discussed in the revised manuscript, the filaments normally start flapping with a frequency comparable to the mode-1 natural frequency, and upon increasing the flow velocity the flapping frequency increases in comparison with the mode-1 natural frequency.

 5) Why the dominant frequency of oscillation increased with increasing Reynolds number?

As discussed in the revised manuscript, the dominant frequency of oscillation increases with Reynolds number because the higher wind speeds provide higher fluid forcing.  

6) The conclusions about the structural response variation with flow velocity are basically observed in the literature (see Paidoussis' work). Could you add another findings from your experiments?

The main objective of this study is to provide data for validation, not to observe or describe new physics. However, in comparison with our previous studies on flexible filaments with circular cross-section, the present filaments with rectangular cross-section do show differences, such as the figure-eight trajectory of the filament free end during limit-cycle oscillations, which was not observed with circular filaments. This is now better described in the revised manuscript.

7) In page 13, you mentioned that the filament dynamics is correlated with Scruton number and reduced velocity. This is correct for the response due to vortex induced vibrations, however at unstable limit cycle oscillations, the excitation mechanism is totally different, it would be fluidelastic instability. Could you explain?

As evident in the definition (Eq. 13 in the revised manuscript), the Scruton number does not depend on the excitation: it only depends on the properties of the structure (linear mass density, damping ratio, and a characteristic dimension which is the filament length in the present case) and on the density of the fluid. The Scruton number can therefore be used anytime the response of a structure exposed to a fluid flow is modulated by its damping, as is the case here.

 8) What is the added contribution of this paper relative to your published papers (1. Flow-Induced Motions of Flexible Filaments Hanging in Cross-Flow/2. Modulation of flexible filaments dynamics due to attachment angle relative to the flow).

As noted above, in comparison with our previous studies on flexible filaments with circular cross-section, the present filaments with rectangular cross-section do show differences, such as the figure-eight trajectory of the filament free end during limit-cycle oscillations, which was not observed with circular filaments. This is now better described in the revised manuscript.

Reviewer 3 Report

Dear Authors, 

The motive of the paper and experimental procedure, devices, and sampling frequency are all excellent.  The outcome of this paper will be a validation tool for several aeroelastic applications.

However, I would like the authors to improve the following points;

  • In the introduction part, there are some experiments done with air as the working fluid, it is ethical if you included and discuss them.
  • In the methodology sections, the authors add some brands and commercial sites such as EnvisionTEC, https://envisiontec.com, a hot-wire anemometer (by Dantec Dynamics, Bristol, UK, www.dantecdynamics.com ; and digital cameras (Panasonic Lumix DMC-FZ200; the technical specification will do the job enough, so no need for the brand names.
  • Furthermore,In the methodology sections, you have not discussed how many numbers of measurements were done for each length and wind speed variations. Also, the statistical approach used for data analysis is not presented. It has to be cleared.
  • The presentation of the results is fine, however, it would be more universal and more useful for many cases if it is presented in normalized and dimensionless form, such as reduced velocity, normalized frequency. Also please try to present other related structural parameters such as the lock-in conditions of the filaments.
  • Finally, Since the main aim of this paper to be a validation tool, the presented raw data is not enough, the should other structural response measurements, maybe as presented in Reference 29 of your manuscript. The presented data for wind speed need descriptions for each row and column it is not clear what is representing.

You have done a very fine job, best wishes!

Respectfully,

Author Response

The motive of the paper and the experimental procedure, devices, and sampling frequency are all excellent. The outcome of this paper will be a validation tool for several aeroelastic applications. However, I would like the authors to improve the following points:

 1) In the introduction part, there are some experiments done with air as the working fluid, it is ethical if you included and discuss them.

The purpose of the Table 1 is not to provide a comprehensive list of all available FSI validation test cases, but only to present a few representative ones to support the discussion, notably the part regarding how the experimental setups are typically realized. This is now better highlighted in the revised manuscript, where Table 1 is introduced with the following sentence:

For illustrative purposes, a non-exhaustive selection of popular experimental FSI test cases is provided in Table 1

2) In the methodology sections, the authors add some brands and commercial sites such as EnvisionTEC, https://envisiontec.com, a hot-wire anemometer (by Dantec Dynamics, Bristol, UK, www.dantecdynamics.com ; and digital cameras (Panasonic Lumix DMC-FZ200; the technical specification will do the job enough, so no need for the brand names.

In previous publications in MDPI journals, we were asked to include those details for the sake of reproducibility.

3) Furthermore, in the methodology sections, you have not discussed how many numbers of measurements were done for each length and wind speed variations. Also, the statistical approach used for data analysis is not presented. It has to be cleared.

Each data point presented in the paper has been generated from averaging one video recording of 15 seconds, corresponding to 3000 frames. Selected data points were repeated, showing good repeatability.

4) The presentation of the results is fine, however, it would be more universal and more useful for many cases if it is presented in normalized and dimensionless form, such as reduced velocity, normalized frequency. Also please try to present other related structural parameters such as the lock-in conditions of the filaments.

In the revised manuscript, we have included a section on the comparative analysis of the response of different filament which also includes plots of the data in normalized form (dimensionless amplitudes and frequencies). The flapping frequencies of the filaments vary continuously with wind speed, without any apparent lock-in.

5) Finally, Since the main aim of this paper to be a validation tool, the presented raw data is not enough, the should other structural response measurements, maybe as presented in Reference 29 of your manuscript. The presented data for wind speed need descriptions for each row and column it is not clear what is representing.

The analysis of the structural response of the filaments provided in the manuscript includes the raw data (amplitude of motion and frequency in horizontal and vertical planes) and a complete characterization of the dynamics of the filament free-end: reconstructed trajectory, the autocorrelation function, and attractor in phase-space. This is more detailed than most literature studies. Regarding the flow field (which was measured in Reference 29), measuring the flow field was not feasible in the present case. Having a flexible structure of small size allows observing large deformations with small Reynolds number, which is presently preferred with FSI validation test cases. The side effect of having a small structure is that the details of the flow also become small, making the visualization of the flow field challenging or impractical (in general, not only for the PIV setup that we had available). This is now better explained in the revised manuscript as follows:

As previously noted, in FSI test cases it is normally preferred to have flexible structures which undergo large deformations with moderate motion frequency whilst interacting with a flow of moderate Reynolds number, thereby avoiding the complications of simulating highly turbulent flows. The small size and high flexibility of the present filaments were instrumental in achieving large displacements and relatively small oscillation frequencies and, at the same time, keep the Reynolds number small. Unfortunately, the details of the flow field scale with the size of the structure, so that the smaller the structure the smaller the space resolution that is needed to faithfully resolve the flow field. In the present case, the requirements for a faithful flow field visualization were beyond our experimental capabilities, and the flow field was therefore not measured.

A note has been included regarding the supplementary material explaining what the columns in the file represent, and under which conditions the data were obtained (sampling frequency, length of recording time).

Reviewer 4 Report

My recommendation will be aligned with comments supplied by Reviewer 1 and I believe the Authors addressed the concerns of this and in the large majority also the other reviewers. It is understood that the authors are proposing to create benchmark data for validation focusing solely on the experimental findings using filaments with rectangular cross-section, albeit measuring the flow field was not feasible in the reported case. 

Round 2

Reviewer 2 Report

Dear authors,

Thank you for considering all comments.

Regards